# A spectrum of modularity in multi-functional gene circuits

Alba Jiménez[1,2], James Cotterell[1,2], Andreea Munteanu[1,2] & James Sharpe[1,2,3,*] iD

## Abstract

A major challenge in systems biology is to understand the relationship between a circuit's structure and its function, but how is this relationship affected if the circuit must perform multiple distinct functions within the same organism? In particular, to what extent do multi-functional circuits contain modules which reflect the different functions? Here, we computationally survey a range of bi-functional circuits which show no simple structural modularity: They can switch between two qualitatively distinct functions, while both functions depend on all genes of the circuit. Our analysis reveals two distinct classes: *hybrid* circuits which overlay two simpler mono-functional sub-circuits within their circuitry, and *emergent* circuits, which do not. In this second class, the bi-functionality emerges from more complex designs which are not fully decomposable into distinct modules and are consequently less intuitive to predict or understand. These non-intuitive emergent circuits are just as robust as their hybrid counterparts, and we therefore suggest that the common bias toward studying modular systems may hinder our understanding of real biological circuits.

**Keywords** decomposability; dynamical mechanism; gene circuits; modularity; multi-functionality

**Subject Categories** Network Biology; Quantitative Biology & Dynamical Systems; Signal Transduction

**Mol Syst Biol. (2017) 13: 925**

## Introduction

A circuit's *structure*, that is, the topology of its interactions, and the biological function it performs do not bear a simple one-to-one relationship (Ingram *et al*, 2006; Payne & Wagner, 2015; Ahnert & Fink, 2016). Indeed, real biological systems show evidence of pleiotropy: Genes, pathways, and even whole circuits often contribute to more than one function. A clear example is found in developmental biology, where just a handful of signaling pathways is essential to the patterning and morphogenesis of many different organs (Pires-daSilva & Sommer, 2003; Carroll *et al*, 2013). Similar phenomena are seen in other types of biological networks, such as neural networks where small sets of connected neurons drive a high diversity of functions in the nervous system (Bargmann & Marder, 2013). So, how is the structure of a circuit influenced if that same circuit must also perform (or contribute to) other distinct functions in the same organism?

A spectrum of hypothetical scenarios can be imagined to explain how two distinct functions are performed in two different tissues (Fig 1). At one extreme, an organism may employ two completely separate circuits: The genes of circuit A are expressed in tissue A and perform function A (and the same for circuit B in tissue B) (Fig 1A). This scenario invokes distinct sets of genes to perform different functions and has been described as *structural modularity* (Hartwell *et al*, 1999; Wagner *et al*, 2007). An alternative scenario is that two modules with distinct tissue-specific functions may share some of their genes (Pires-daSilva & Sommer, 2003; Carroll *et al*, 2013). The shared genes will be expressed in both tissues, while the remaining genes may only be expressed in one tissue or the other (Fig 1B and C). Although the two modules still function separately, their circuitry overlaps, that is, there is a loss of structural modularity. This theoretical spectrum of scenarios helps us to define what we mean by *multi-functionality*: The scenario in Fig 1A is not multi-functional at all, and the degree of multi-functionality increases as we progress to the right until the complete overlap of modules in Fig 1D.

Studies of multi-functionality have considered this concept in a variety of different ways. The common underlying theme is that some kind of "change" to the circuit leads to a different "outcome". However, different scenarios can lead to slightly different interpretations of the meaning of multi-functionality. Although the following categories are not always mutually exclusive, we propose them as a useful framework for comparing and contrasting alternative situations:

1   *Mutually compatible functions*. Some circuits have been studied which perform more than one qualitatively distinct function, but whose functions are simultaneously compatible—they do not in fact require any change to the circuit. For example, Ten Tusscher & Hogeweg (2011) describe multi-cellular pattern-forming circuits which can perform two functions: segment a field of cells and also provide the segments with different expression identities. Since the spatial expression

1   EMBL-CRG Systems Biology Research Unit, Centre for Genomic Regulation, The Barcelona Institute of Science and Technology, Barcelona, Spain
2   Universitat Pompeu Fabra (UPF), Barcelona, Spain
3   Institució Catalana de Recerca i Estudis Avançats (ICREA), Barcelona, Spain
    *Corresponding author. Tel: +34 93 316 0098; E-mail: james.sharpe@crg.eu

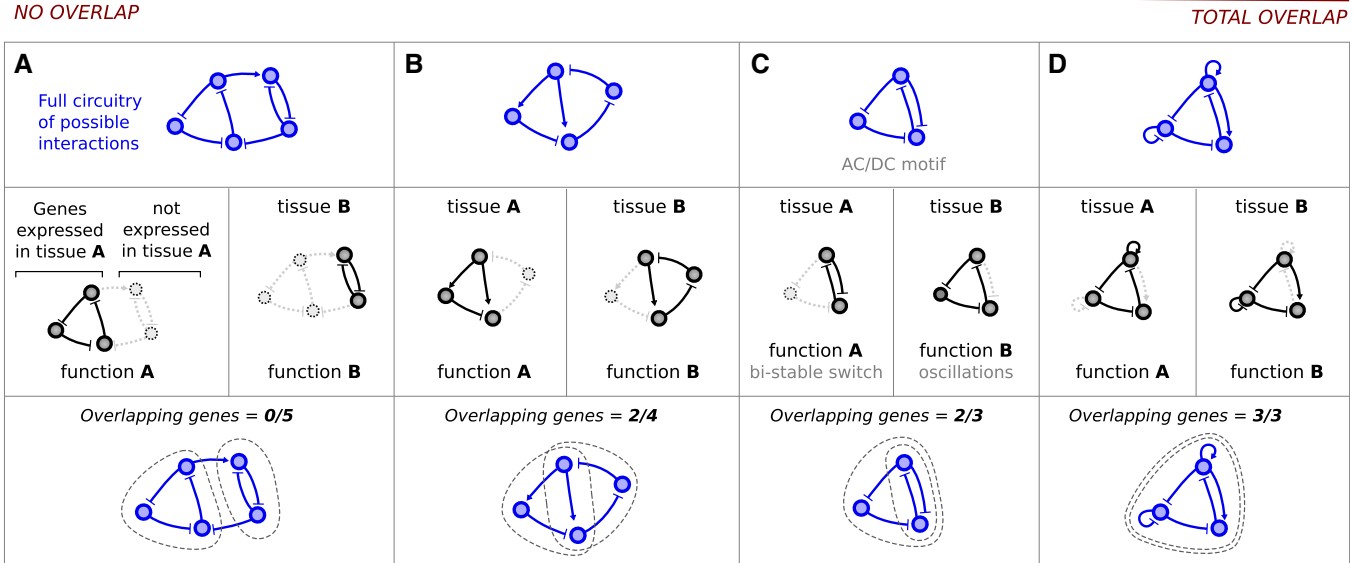

**Figure 1. Defining a multi-functional circuit.**

A In order to achieve multiple functions it has been proposed that circuits can be *structurally modular*, i.e they allocate distinct highly interconnected and non-overlapping sets of genes to each individual function (Di Ferdinando *et al*, 2001; Solé & Valverde, 2008; Clune *et al*, 2013; Ellefsen *et al*, 2015). In this scenario modules do not overlap.

B, C Partial module overlap (Panovska-Griffiths *et al*, 2013; Sorrells *et al*, 2015). The AC/DC circuit is able to alternate between distinct behaviors upon a change in the strength of its gene interactions. This circuit is formed by the superimposition of two distinct modules, a mutual inhibition motif and a repressilator motif, that combine under the same topology. As the strength of specific repressive interactions is adjusted, the AC/DC circuit switches between two distinct dynamical behaviors, that is, a bistable switch or oscillatory behavior, using the mutual inhibition and repressilator circuits, respectively.

D Hypothetical scenario describing a complete module overlap: The same collection of interacting genes is essential to both functions.

patterns are different from each other (one is periodic, and the other is aperiodic giving each segment a different identity), thus the functions have to use at least partially different sets of genes to express the different spatial patterns. This is not the type of multi-functionality we seek in the current study, as we are looking for a minimal network where the two functions depend on the exact same set of genes.

2 *Multi-stable circuits*. Some studies have considered multi-stability as a type of multi-functionality. Payne and Wagner (2013, 2015) studied small Boolean circuits and considered each possible stable attractor as a distinct function. Multiple attractors in the phase portraits of continuous dynamical circuits have also often been used to represent alternative cell types in models of cell fate choice—for example, Slack, 1991; Huang *et al*, 2007; Graf & Enver, 2009; Corson & Siggia, 2012; Furusawa & Kaneko, 2012;. However, the different end-states in these examples (whether Boolean or continuous) are typically stable point attractors, which do not correspond to qualitatively distinct dynamical behaviors (although cyclic attractors are also possible in these systems). Indeed, in many other contexts, a circuit which "classifies" many possible input states, or initial conditions, into a few final states, is considered to have a single "decision-making" function. Good examples include pattern-recognition circuits (in the field of neural networks; Sussillo & Abbott, 2009) and bi-fan motifs, in which a single motif function is similar to a Boolean "truth table",

mapping all possible combinations of input states to few distinct output states (Ingram *et al*, 2006).

3 *Altering circuit structure*. Other studies have explored cases in which the switch in function requires an explicit change in circuit structure. Kashtan & Alon (2005), Kashtan *et al* (2009) found that Boolean circuits which evolved in an environment with alternating fitness functions contained distinct modules. However, these modules did not reflect the two alternative functions of the whole circuit—instead they reflected two "subproblems" which were common to both of the functions—a feature the authors termed "modularly varying goals". (The modules were therefore providing mutually compatible sub-functions (an example of point 1 above) rather than alternative functions). Also, since function-switching depended on "mutating" the structure of the circuit—not just changing parameter values—each given circuit was in fact mono-functional.

4 *Multi-functional circuits*. Many other studies have found simple circuits which can switch between distinct dynamical behaviors without requiring a change in structure—just a change in parameter values. Turing circuits can switch between forming spots or stripes (Meinhardt, 1982; Lin *et al*, 2009), although these two outcomes are qualitatively very similar (both being periodic patterns in space). Other circuits display phase transitions between two qualitatively distinct behaviors, such as oscillatory dynamics and bi-stability

 

(François & Hakim, 2005; Rouault & Hakim, 2012; Panovska-Griffiths *et al*, 2013). These examples tend to be quite simple dynamical systems; nevertheless, this type of multi-functionality is closer in spirit to the general biological phenomenon of pleiotropy because the alternative functions are not simply alternative decision states (which need not be qualitatively distinct), but instead they directly embody the distinct dynamical behaviors of two different biological functions.

Although these previous studies have provided important insights into questions of multi-functionality, here we seek to focus in a more detailed manner on the question of modularity or *decomposability*. Specifically, which nodes and which links of a multi-functional circuit are involved in each of its functions? Can we always decompose a multi-functional circuit into the distinct sub-modules that underlie each function? And can we understand the structure-to-function relationship in terms of both *decomposable structure* and *decomposable dynamics*? Additionally, we designed our study to go beyond the analysis of just one or two chosen circuits, and instead perform a systematic survey across a given class of circuits, so that more general conclusions can be drawn. Finally, to enhance the qualitative distinction between alternative functions, we chose to go beyond temporal (single cell) dynamics and to explore behaviors which show both temporal and spatial dynamics in a multi-cellular field.

To address these questions, we take inspiration from the broadly studied Notch-Delta paracrine signaling pathway, which can exhibit two mutually exclusive, and qualitatively distinct behaviors: *lateral induction* or *lateral inhibition* (Lewis, 1998; Fig 2A). The first of these, lateral induction, is the process by which a cell signals to its neighbors to adopt the same gene expression state. It results in a dynamic, progressive spreading of this state across the tissue, like a wave propagating from cell to cell resulting finally in a continuous domain of cells expressing the same genes. In contrast, in lateral inhibition cells inhibit their neighbors from adopting the same state, leading to a "salt and pepper" pattern of cells in alternating differentiation states. Unlike lateral induction, this behavior does not depend on a progressive spread across the field—the pattern may appear simultaneously everywhere. Both the molecular details of how Notch and Delta interact (Collier *et al*, 1996; Lewis, 1996; de Celis *et al*, 1997; Panin *et al*, 1997; Formosa-Jordan & Ibanes, 2009), and the variety of patterns they can achieve (Palau-Ortin *et al*, 2015), such as boundary formation (Huppert *et al*, 1997) or synchronization of oscillations (Horikawa *et al*, 2006), are more complex than the model we explore here. In our case, the goal of this model is not to understand the details of real Notch-Delta signaling, but instead to provide an abstract but biologically relevant model to explore the concept of multi-functionality.

# Results

## The model

To explore multi-functionality in paracrine signaling circuits, we developed a simple model of direct cell–cell communication (similar to Salazar-Ciudad *et al*, 2000; Plahte, 2001), which considers both inter- and intra-cellular gene regulation in a one-dimensional spatial

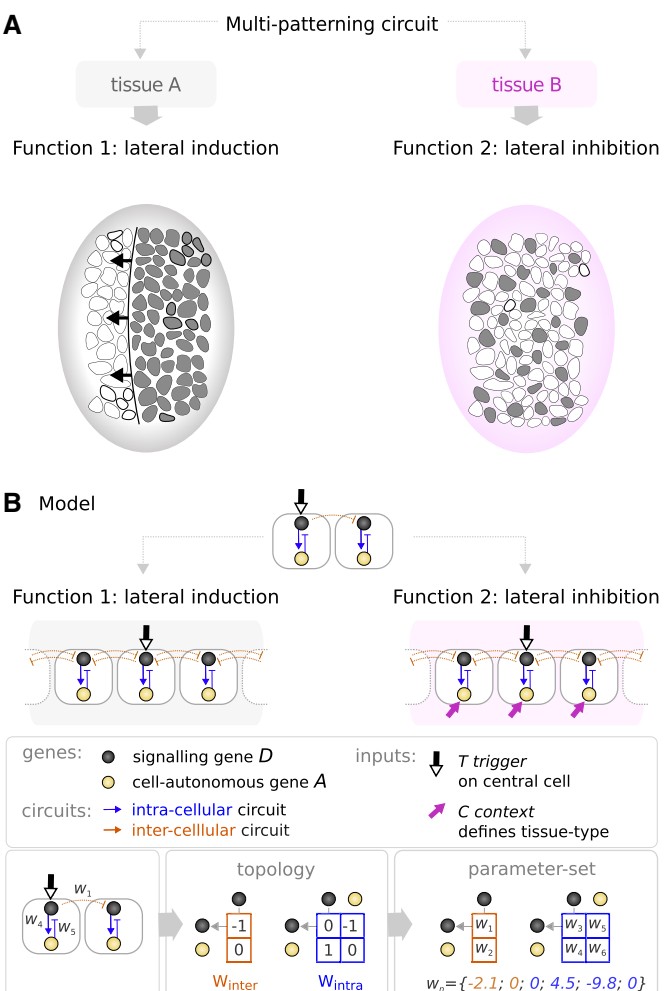

**Figure 2.  Defining a multi-patterning circuit.**

A   We explore multi-functional circuits capable of two qualitatively distinct multi-cellular patterns: lateral induction and lateral inhibition. Analogous to biological processes such as the progression of the morphogenetic furrow in *Drosophila* (Sato *et al*, 2013), lateral induction leads to the propagation in time and space of a given gene expression state. In contrast, lateral inhibition describes processes such as neurogenesis, where a fine-grained pattern of alternating cell fates is formed (Daudet & Lewis, 2005; Petrovic *et al*, 2014).

B   While genes (represented by black and yellow nodes) interact identically in both tissues/contexts, an external input signal termed the context signal *C* allows the circuit to switch between functions. The context signal (pink arrow) affects the basal expression level of one of the genes in every cell of the tissue. A circuit achieves lateral induction when it causes a progressive spread of expression from trigger *T* (thick black and white arrow) which is received by the central cell of the tissue. A circuit achieves lateral inhibition when it causes consecutive cells to be in alternating gene expression states. In subsequent figures, we use a simplified 2-cell representation where, for simplicity, the inter-cellular circuit is only shown in one direction (from the first cell to the second).

system comprising 33 cells (Fig 2B and Box 1). To restrict the search to the simplest "minimal" multi-functional circuits, we consider two-gene circuits where only one signaling gene (the black node in Fig 2B) is able to regulate expression of genes in the neighboring cells. This signaling gene is labeled *D* (black), as it can be seen to

**Box 1:   The gene regulatory model**

The model describes how the concentration $g_{ij}$ of the $i^{th}$ gene in the $j^{th}$ cell changes with time:

$$\frac{dg_{ij}}{dt} = \chi\left[\Phi\left[\sum_{l=1}^{2} W_{inter}^{li} g_{lj} + W_{intra}^{li} g_{lj} + T + C\right]\right] + \lambda g_{ij} + \eta(t) g_{ij} \tag{1}$$

where $\chi(x)$ is the Heaviside function to prevent negative gene product production rates, $W_{inter}$ and $W_{intra}$ are matrices containing the strengths of gene interactions (Fig 2B), $T$ is the trigger signal received by one of the genes in the central cell and equal to 1, $C$ is the context signal received by one of the genes in every cell of the tissue and defines tissue-specificity (when $C = 0$ the circuit is embedded in tissue A and when $C = 1$ the circuit is in tissue B), $\lambda$ is the decay rate equal to 0.05 and $\eta(t)$ is a noise term, which adds uniformly distributed fluctuations ($\pm$ 1%) to the concentration of every gene in every cell at every time step. $\phi(x)$ is the regulatory function that describes the rate of change of a gene level in response to its various regulatory inputs. It represents the real process of gene regulation in which multiple transcription factors bind specific cis-regulatory regions (Setty et al, 2003; Ben-Tabou de-Leon & Davidson, 2009). Here, we employed the commonly used sigmoid function with the following particularity: We included a parameter $\alpha_i$ which allows each gene to independently adopt a qualitatively distinct regulatory behavior with respect to their inputs: either showing a constitutive expression, that is, being transcribed in the absence of input values or despite negative inputs (e.g., $\alpha_i = -60$), or being dependent on lower ($\alpha_i = 15$) or higher ($\alpha_i = 60$) input value. $\beta$ controls the steepness of the function.

$$\phi(x) = \frac{1}{1 + \exp^{\alpha_i - \beta x}} \tag{2}$$

We use reflective or zero-flux boundary conditions that do not allow any diffusion in or out of the system, therefore modeling the system as isolated from other tissues. The simulation starts with every gene in every cell set to have a concentration of 0.1. The two types of external input signals $T$ and $C$ are kept constant throughout the simulation.

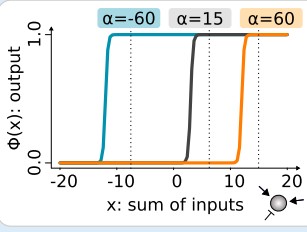

represent the Notch-Delta signaling pathway, while the second gene is cell-autonomous and is labeled A (yellow).

We had to define objective functions to determine whether a given circuit achieves lateral induction, lateral inhibition or neither of these behaviors. The process of lateral induction cannot be assessed by analyzing the end-point of a simulation. The correct end-point is a uniform gene expression pattern, but this observation alone is ambiguous—much simpler processes than lateral induction could also result in this end-point pattern (e.g., a circuit with a constant positive input to a gene). Lateral induction is thus intrinsically a dynamical process in which a given cell state propagates through the field of cells resulting in a domino-effect—each newly activated cell inducing the next one. Identifying this dynamical

process therefore depends on assessing the state of the simulation at multiple time-points. To observe this propagatory wave, we therefore set up all simulations with a pre-defined molecular trigger $T$ at a specific cell in the middle of the field, which acts as an external input to one of the genes of the circuit. In contrast, the scoring of lateral inhibition does not depend on the dynamics of pattern formation but only on the final pattern. Once the pattern has reached equilibrium, lateral inhibition is scored by counting the number of consecutive cell state changes from high to low expression of the patterning gene. We allow for imperfections and thus consider successful patterns that show more than 13 of those cell-state switches from high to low expression. In a 33 cell tissue, a "perfect" lateral inhibition pattern would count 16 of those switches. Although the trigger $T$ is not strictly necessary for the lateral inhibition case, we include it so that the conditions are identical in all simulations (Appendix Fig S1).

### Finding minimal circuits for either lateral induction or lateral inhibition

We chose to search for multi-functional circuits in a two-step process. Since bi-functional circuits may be direct combinations of two simple mono-functional circuits, we first focused on identifying all the minimal circuits able to achieve either induction or inhibition alone.

We performed an exhaustive and unbiased search through gene circuit space by enumerating all the 1,200 possible two-gene topologies (see Materials and Methods) and subsequently sampling large numbers of parameter sets ($10^7$) for simulating each topology. This type of systematic exploration of circuit space has been used previously to explore the design space of small circuits for single functions such as gradient interpretation (Cotterell & Sharpe, 2010; Schaerli et al, 2014), somite formation (Cotterell et al, 2015), biochemical adaptation (Ma et al, 2009), or cell polarization (Chau et al, 2012).

Using the objective functions described above, 584 topologies were found to achieve lateral induction, and 632 lateral inhibition. These are displayed in a complexity atlas (Cotterell & Sharpe, 2010; Cotterell et al, 2015; Jiménez et al, 2015) in blue or red, respectively (Fig 3A). Similar to a neutral network (Schuster et al, 1994), a complexity atlas is a metagraph that comprises all circuits achieving a given function, and is constructed with the following two rules: (i) Gene circuit topologies (nodes of the atlas) are directly connected (by edges) if they differ by a single gene interaction, that is, topologies differ by the gain or removal of one regulatory interaction. (ii) Topologies are ordered along the vertical axis with respect to their complexity (which we define as number of regulatory links) with complexity increasing upwards. This way, the simplest successful circuits—the minimal topologies —appear at the bottom of stalactite-like structures in the atlas (Cotterell & Sharpe, 2010). A number of topologies were capable of both behaviors (lateral induction or lateral inhibition) depending on the parameter values, but before studying these potentially bi-functional designs (next section), we chose to explore in detail the various minimal mechanisms which perform either lateral induction or lateral inhibition separately.

From the tips of the stalactite-like structures in the atlas (Fig 3A), we obtained the minimal circuits to achieve lateral induction (D0–D5) or lateral inhibition (H0–H5) (Fig 3B and C). Within each function, core circuits could be classified according to the

                    

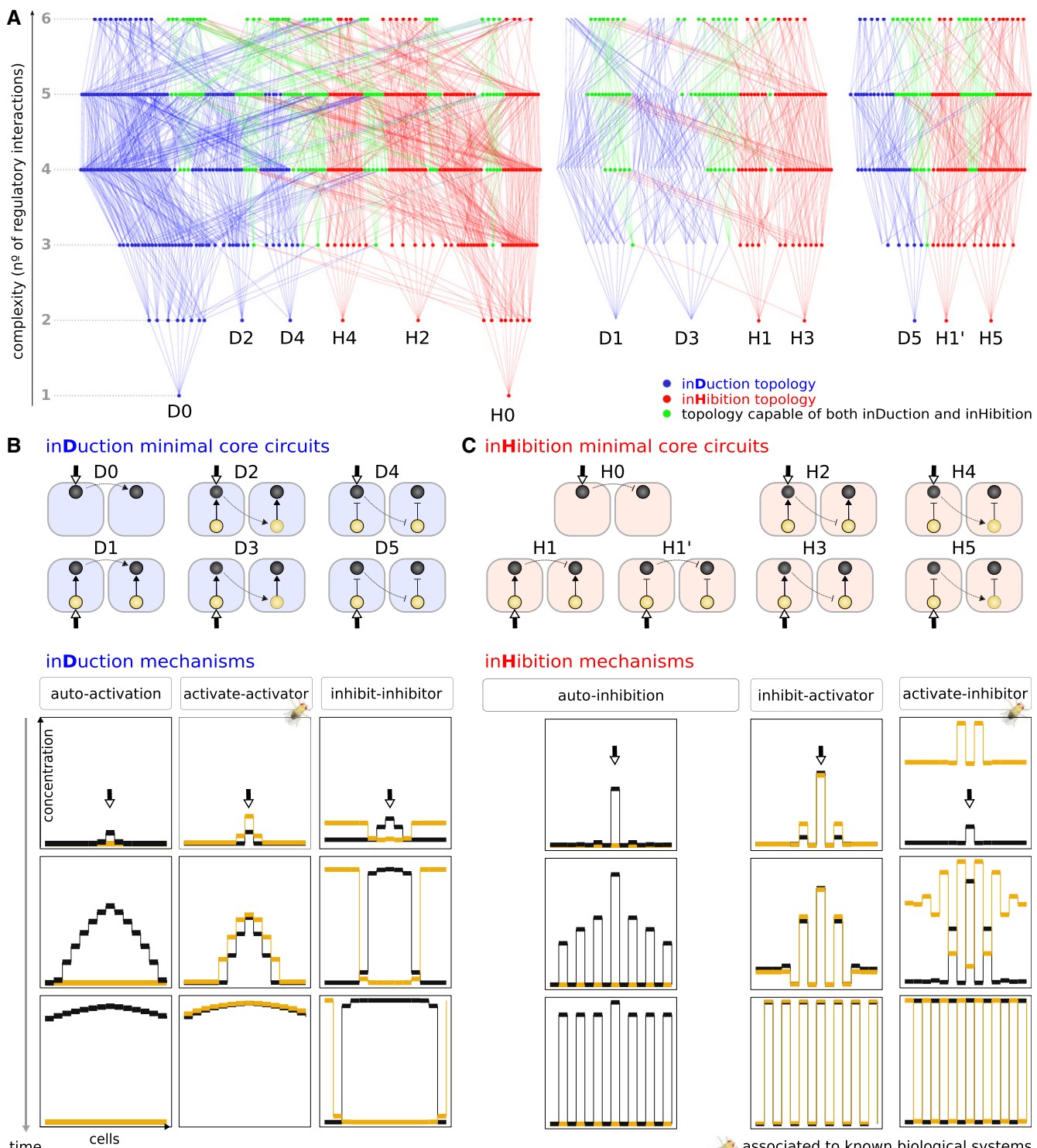

**Figure 3. Dynamical mechanisms of lateral induction and lateral inhibition.**

A      Color-coded complexity atlas that contains all two-gene circuits able to achieve lateral induction or lateral inhibition. Nodes are circuit topologies and edges link those with a single-topological change (addition or removal of a gene interaction). Topologies (nodes) are colored according to their function: Blue and red exclusively hold circuits capable of induction or inhibition, respectively. Green topologies are capable of both induction and inhibition (see Fig 4). The atlas layout, where topologies are ordered according to their number of regulatory links, reveals the core motifs at the tip of stalactites.

B, C    Minimal core circuits to achieve induction (D0–D5) or inhibition (H0–H5) are classified into three distinct mechanisms for each function. Alternative mechanisms correspond to distinct spatiotemporal courses of gene expression to achieve a given function. The dynamical strategy of each mechanism is captured in the unique final profile. While simulations take places on a one-dimensional row of 33 cells, for increased clarity, most graphic representations show 15 cells.

dynamical strategy they use, that is, their *dynamical mechanism*. While circuits D0–D5 all lead to a spread of a given cell state through the tissue (induction), they do so with distinct time-courses of gene expression, thus using alternative dynamical mechanisms to achieve this behavior (Fig 3B, Appendix Figs S2A and S3A and B). The simplest mechanism, that of *auto-activation* (D0 and D1), leads to the autonomous expansion of the signaling gene. Instead, the mechanism *activate-activator* (D2 and D3) orchestrates a synchronous in-phase expansion of both genes, while *inhibit-inhibitor* (D4 and D5) leads an out-of-phase expansion of both genes as their expression profiles are complementary, that is, the signaling gene (black node, *D*) is able to spread by lowering expression of its inhibitor (yellow node, *A*). In a similar way, three alternative dynamical mechanisms were found for inhibition (Fig 3C, Appendix Figs S2B and S3A and C).

These initial findings about the simplest mono-functional circuits propose a handful of alternative mechanisms likely to be used in real biological systems. Interestingly, among the mechanisms found, some have been proposed to orchestrate specific biological processes (marked with a *Drosophila* symbol in Fig 3B and C): *activate-activator* controls dorsoventral boundary formation in the *Drosophila* wing (de Celis & Bray, 1997; Panin *et al*, 1997) while *activate-inhibitor* is associated to processes such as neurogenesis in vertebrates, flies, and worms (Lewis, 1996), shaping *Drosophila* wing vein morphogenesis (Huppert *et al*, 1997) or synchronizing oscillations during somitogenesis (Horikawa *et al*, 2006). More importantly, those mechanisms not known to be used in real biological systems constitute potential candidate designs to achieve lateral induction or lateral inhibition in other organisms.

## Strongly bi-functional circuits: Maximal module overlap and minimal structural change

Our complexity atlas revealed that some topologies ($N = 258$) are compatible with both functions (green nodes in Fig 3A). From the finding of these bi-functional circuits, two observations follow.

First, while 58% of all topologies can perform a single function, only 21% of them are able to perform both functions. This observation supports the finding of previous work with Boolean circuits (Payne & Wagner, 2013, 2015) which showed that there are fewer multi-functional circuits than mono-functional ones (albeit using a distinct definition of multi-functionality, see Introduction). This supports the intuitive idea that multi-functionality constrains circuit structure. Second, distinct bi-functional topologies have different likelihoods to implement lateral induction or lateral inhibition. The probability of achieving each function is given by the proportion of the $10^7$ sampled parameters that yield induction or inhibition. We thus picture each bi-functional topology as a pie chart showing the proportion of parameter space that gave each function (Appendix Fig S4). We notice that most bi-functional topologies are strongly biased toward one of the functions, that is, *specialized* (Macía *et al*, 2009)—pie charts which appear almost entirely blue or entirely red, while fewer topologies hold similar probabilities to implement each function, that is, *flexible* (Macía et al, 2009).

For each of these bi-functional topologies, their behavior can be switched from lateral induction to lateral inhibition by changing one or more of the weighting parameters ($w_n$) that define the strength of the regulatory interactions. Two examples of such circuits are shown in Fig 4. In a real organism, the strength of any regulatory link could be modulated in a tissue-specific manner, by controlling the expression levels of a co-factor protein (Narita & Rijli, 2009). However, we wished to explore further the minimal change in the circuit able to cause a switch in function. Could we maintain the innate structure of the circuit unaltered—without any changes to the topology or the weights of the regulatory interactions? That is, switch the function while the genes still interact in the exact same manner (with the same $w_n$ parameters)? The minimal influence on the circuit we could imagine was to allow the background expression level of just one of the constituent genes to be different in the two tissues. We thus introduced a second external input signal termed the context signal *C*. This signal defines two distinct tissue

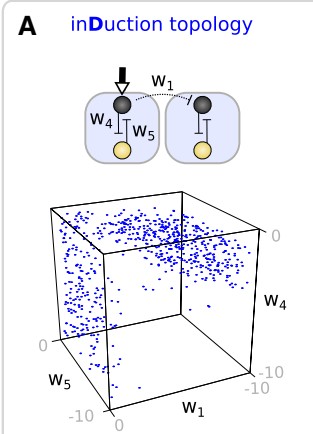
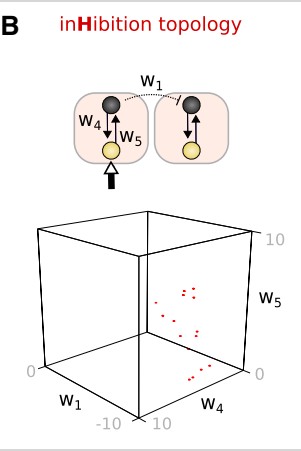
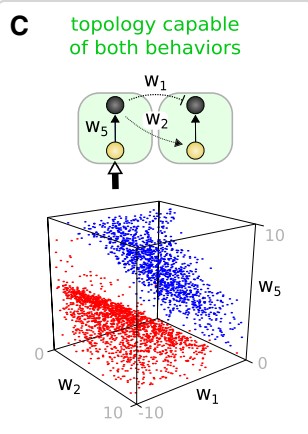
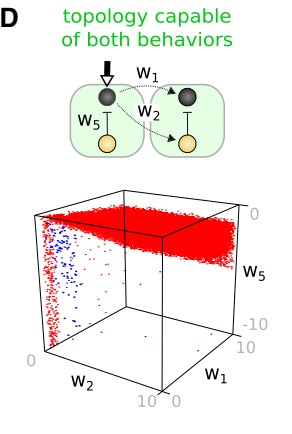

**Figure 4.  Parameter spaces for mono-functional and bi-functional circuits.**

A–D   Unlike mono-functional topologies (A, B), green topologies (C, D) (see Fig 3A) are capable of induction and inhibition depending on the values of their gene interactions. (C, D) Each function occupies a distinct region in parameter space.

environments: a first environment in which the basal expression level of one of the genes is unaltered (tissue A, $C = 0$) and a second environment in which that same gene will have a higher basal expression level homogeneously throughout the tissue (tissue B, $C = 1$). A successful *strong* bi-functional circuit would be one which performs lateral induction in one tissue and lateral inhibition in the other (or *vice versa*). In order to identify such circuits, we considered all previous simulations to have taken place in the first environment (tissue A), as $C$ was zero in these cases. Using the same objective functions as before, we re-simulated this initial pool of circuits in the second environment (tissue B) as we add the context

signal (to each of the genes in turn) and selected those able to switch from their original function to the alternative one. We identified 72 different topologies capable of strong bi-functional behavior (1,130 parameter sets in total). Highlighting these topologies as black nodes and edges in the complexity atlas revealed that they formed a new restricted set of stalactites (Fig 5A, Appendix Fig S5A). As before, the minimal bi-functional circuits are revealed at the bottom of these new stalactites (full list given in Appendix Fig S5B). We have thus identified a collection of strongly bi-functional circuits (referred in the text from now on as simply bi-functional): Not only do both functions depend on both genes, but the switch in

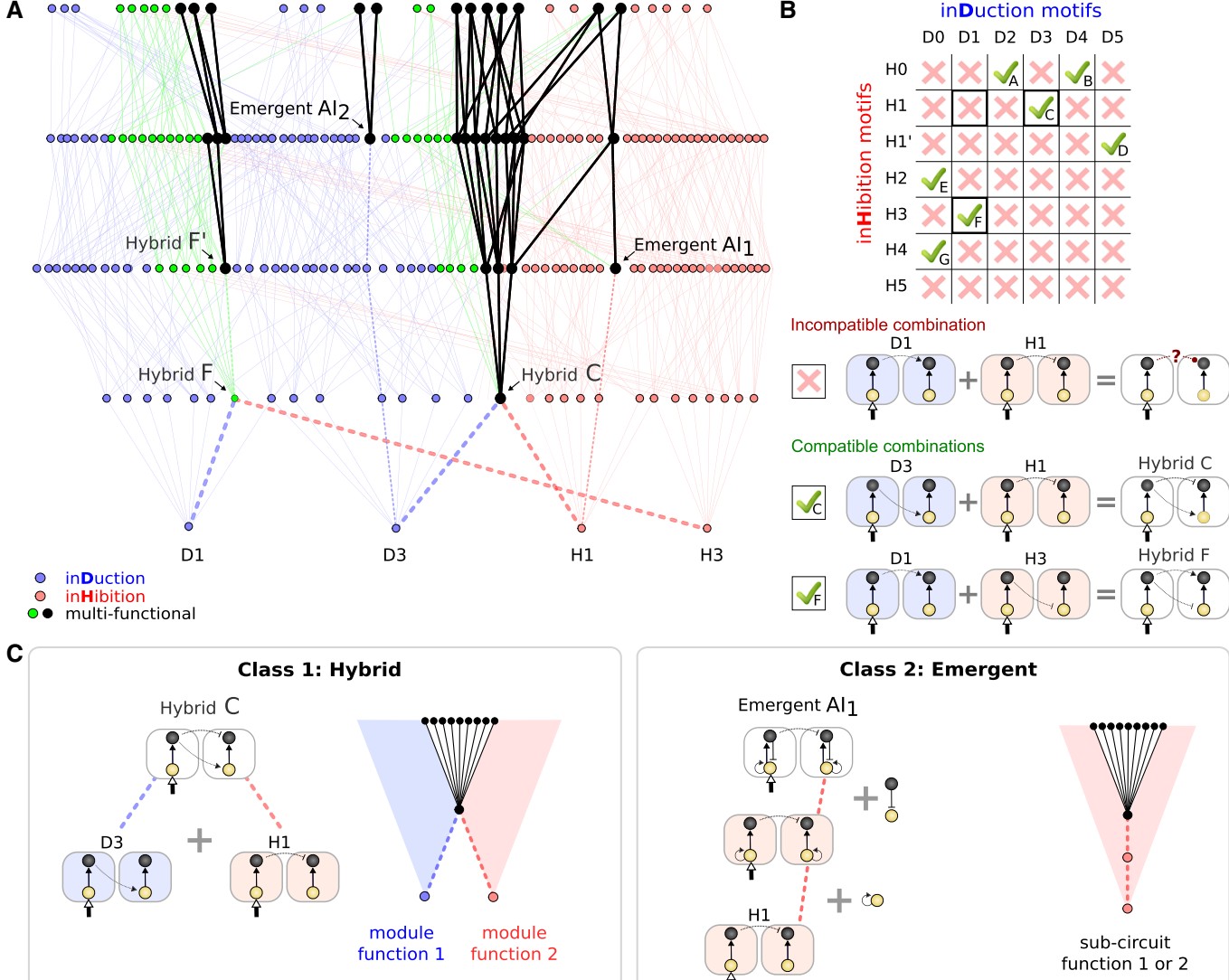

**Figure 5.  Two classes of multi-functional circuits.**

A    Subregion of the complete atlas (Appendix Fig S5A) where strong multi-functional circuits are shown in black. Of the 13 bi-functional core circuits (Appendix Fig S5A and B), four are shown here.

B    Compatible combinations of two core induction and inhibition circuits are candidates to multi-functionality, labeled A to G.

C    Multi-functional motifs show distinct modular properties. Hybrids are composed of two separable modules, or sub-circuits, while emergent circuits cannot be decomposed into distinct sub-circuits. As such, hybrid circuits visually appear as the sum of two induction and inhibition circuits—the union of two mono-functional stalactites—while emergent circuits "emerge" at higher levels of complexity within a stalactite.

function is achieved just by changing the background expression level of one of the genes (in other words: maximal circuit overlap, with minimal structural change between functions).

## Two classes of multi-functional circuits: hybrid and emergent

As defined in the introduction, these circuits are not structurally modular, that is, their functional modules are tightly overlapped. But to what extent can these circuits nevertheless be understood as the union of simpler circuits? In other words, to what extent can they be decomposed into two basic mono-functional circuits—each responsible for one of the functions?

Figure 3 lists all the minimal mono-functional circuits for our two functions (lateral induction and lateral inhibition). We first had to consider which pairs of these simple circuits are *topologically compatible* to be combined into potentially bi-functional circuits. Figure 5B shows that not all combinations are possible. For example, D1 and H1 cannot be combined, because they have opposite signs for the same regulatory interaction. On this basis we found that, of the 42 hypothetical combinations, seven are topologically possible, which we label A to G (Fig 5B, Appendix Fig S6).

We could now examine which of the compatible *hybrid* circuits (A–G) appeared in the complexity atlas, that is, which are successfully bi-functional (black nodes in Fig 5A, Appendix Fig S5A). Most, but not all, of the simple hybrids gave bi-functional behavior. For example, C (which is the hybrid of D3 and H1) does produce a bi-functional circuit, while F does not (Appendix Fig S5A). However, for all non-bi-functional minimal hybrids, the addition of an extra regulatory link was able to render the circuit successfully bi-functional. For example, the addition of positive auto-regulation to the cell-autonomous gene of F produces a bi-functional circuit which we label F'. All these modified hybrids are found one level higher in the atlas and are listed as A' to G' (Appendix Fig S5A and B). Thus, our first general conclusion is that many bi-functional circuits can indeed be decomposed into their simpler constituent mono-functional "building blocks". The decomposability of hybrids is graphically illustrated within the structure of the complexity atlas itself (Fig 5C). Hybrid circuits can be found at the position where two mono-functional stalactites meet (e.g., in Fig 5A stalactites D3 and H1 fuse to create hybrid C, while in Appendix Fig S5A stalactites D0 and H4 fuse to create hybrid G).

However, examination of the complexity atlas also revealed a very different class of circuits, which do not fit this picture. Some of the successful bi-functional circuits (e.g., $AI_1$ and $AI_2$ in Fig 5A or *Pattern-Convertor* in Appendix Fig S7) are not decomposable into two simpler mono-functional circuits. This can be appreciated both in terms of their circuit structure, and their position within the atlas, which we illustrate here by describing $AI_1$ (so named because it contains an *Activator-Inhibitor* feedback loop within it). Structurally, $AI_1$ contains the H1 module within it, and this is responsible for its ability to perform lateral inhibition. However, the second function— lateral induction—cannot be explained by a module or sub-set within $AI_1$. Specifically, none of the minimal lateral induction circuits (D0–D5) can be found within it. Instead, the lateral induction functionality only arises when we consider the whole circuit at a higher level of circuit complexity, and is dependent on the whole bi-functional circuit—no sub-set of $AI_1$ is capable of lateral induction. To appreciate this in more detail (Fig 5C), if we consider the

mono-functional lateral inhibition circuit H1, two extra regulatory interactions must be added to create the bi-functional $AI_1$ (an auto-activation in the yellow gene and a repression from the black to the yellow gene) but neither of these additions re-creates any of the minimal circuits D0–D5.

We call this second class *emergent* bi-functional circuits, because one of the two functions only emerges at higher level of complexity, when the full bi-functional circuit is complete. Again our complexity atlas proves a useful way to visualize the mechanistic relationships. Unlike hybrid circuits, which are found at the union of two mono-functional stalactites, emergent circuits always arise inside a single stalactite. The position of emergence is a few steps up from the lowest/minimal point of the stalactite, and the path from one to the other (dashed lines in Fig 5A and C) represents the addition of the extra regulatory links which must be added to the minimal mono-functional circuit.

## The dynamics and decomposability of hybrid bi-functional circuits

Our analysis above proposed that two classes exist for bi-functional circuits: *hybrids* which are clearly decomposable into two modules or sub-circuits, each of which is responsible for one of the functions, and more complex *emergent* circuits in which one of the functions cannot be explained by a sub-module and thus depends on all regulatory links of the full circuit. To explore this result in more detail, we wished to go beyond the structural analysis described above and explore whether the same distinction could be found in terms of circuit *dynamics*. As previous studies have proposed (Slack, 1991; Huang *et al*, 2007; Macía *et al*, 2009; Corson & Siggia, 2012; Rouault & Hakim, 2012), a geometric analysis of a system's phase space can help us understand how this system can give rise to different cell states. Indeed, a circuit's design (structure and parameter values) encodes the shape of the nullclines which in turn determine— through their crossing—the steady states of the system, basins of attraction and separatrices (Strogatz, 2014).

This "dynamical systems" analysis of hybrid and emergent circuits aims to ask: Are the two different classes of circuits distinguishable from the geometry of their phase space? In this section we first tackled hybrid circuits, to explore whether their dynamics are a simple composition of the underlying mono-functional modules. Afterward, in the subsequent section, we address emergent circuits.

To address the *dynamical decomposability* of hybrid circuits, we independently needed to understand how the two functions (lateral induction and lateral inhibition) are implemented as distinct dynamical mechanisms. As this study focuses on spatial patterning circuits, our phase portraits must represent the states of more than one cell. We thus chose a simplified 2-cell model (Box 2) to describe how the states of two neighboring cells affect each other. This simplified model has four state variables: $D_{c1}$, $A_{c1}$, $D_{c2}$, $A_{c2}$, (the concentrations of the signaling and cell-autonomous genes in *cell 1* and *cell 2*). To keep the phase portraits 2-dimensional we describe the dynamics of the system using just one variable per cell, and we chose $D$ since it represents the essential signaling event. Thus, unlike many previous phase portraits where attractors represent alternative differentiation states of single cells (Slack, 1991; Huang *et al*, 2007; Corson & Siggia, 2012), the steady states in our phase portraits represent different multi-cellular patterns.

We draw the phase portraits of all minimal induction (D0–D5) and inhibition (H0–H5) circuits to observe the various arrangements of steady states and nullclines (Appendix Fig S8). The general features of the geometry of phase space that characterize induction and inhibition functions are illustrated in Box 2.

From the general dynamics of lateral induction and lateral inhibition (Box 2), we can now address the question of how hybrid circuits work and we chose the circuit hybrid C (Fig 5) as a concrete case to study. We compare the key dynamical features of hybrid C

for each of the two functions to those of its corresponding mono-functional modules (D3 and H1) (Fig 6). The collection of steady states of hybrid C as it achieves induction (tissue B, $C = 1$) is exactly the same as for its underlying induction module D3. Both phase portraits show two attractors, at positions low–low ($\theta_1$) and high–high ($\theta_2$), and upon receiving a signal from a neighbor cell, the lower steady states annihilates and the system shifts up to the remaining attractor $\theta_2$ (Bifurcation 1 Fig 6A). The arrangement of steady states and the annihilation we observe are characteristic of

---

**Box 2:   Dynamics of lateral induction and lateral inhibition**

From the general gene regulatory model in Box 1, we can write the equations of a simplified two-cell model for any given circuit, for example:

$$\frac{dD_{c1}}{dt} = \frac{1}{1 + \exp^{z_D - \beta(w_5 A_{c1} + w_1 D_{c2})}} - \lambda D_{c1} \tag{3}$$

$$\frac{dA_{c1}}{dt} = \frac{1}{1 + \exp^{z_A - \beta(T + w_2 D_{c2})}} - \lambda A_{c1} \tag{4}$$

$$\frac{dD_{c2}}{dt} = \frac{1}{1 + \exp^{z_D - \beta(w_5 A_{c2} + w_1 D_{c1})}} - \lambda D_{c2} \tag{5}$$

$$\frac{dA_{c2}}{dt} = \frac{1}{1 + \exp^{z_A - \beta(w_2 D_{c1})}} - \lambda A_{c2} \tag{6}$$

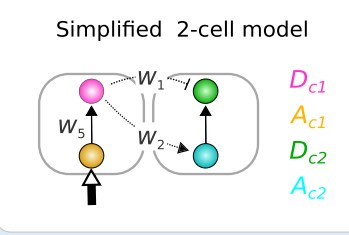

Simplified  2-cell model

$D_{c1}$
$A_{c1}$
$D_{c2}$
$A_{c2}$

Using this simplified model, we draw the instantaneous phase portraits (Verd *et al*, 2014) of D5 and H5 as concrete examples to study the dynamics of induction and inhibition, respectively. First, we explore how lateral induction is

performed—that is, how a wave of gene expression progresses across the field. The dynamical requirements for a propagatory wave are that each cell should stably maintain low expression until the wave of induction reaches it. Once a cell receives an inductive signal from its neighbor, it them moves up to a new stable state of high expression. This is exactly what we observe as D5 performs induction. When the pair of cells has no external input (e.g., the initial state of cells near the edge of the field) the nullclines for $D_{c1}$ and $D_{c2}$ cross at three positions, and the system is indeed bi-stable (two stable steady states, or *attractors*, with an unstable steady state in between). The initial conditions of the simulation have low levels of $D$ in every cell, so the simplified 2-cell system remains stably at $\theta_1$ (the attractor where $D_{c1}$ is low and $D_{c2}$ is also low, which we will call low–low). When *cell 1* receives the inductive signal (either from the trigger $T$, or from a neighboring cell which is propagating the wave) this effectively shifts the nullcline for $D_{c1}$ (see zoomed region of the phase portrait on the right), such that both steady states in this region disappear. In dynamical systems theory the steady states are said to have annihilated each other—an event called a *bifurcation* (Strogatz, 2014). There is only one remaining steady state, $\theta_2$, which is high expression for both $D_{c1}$ and $D_{c2}$ (high–high), so both cells move up to this new state. This sequence of events holds for every pair of cells in the field, and so the inductive wave propagates in a controlled manner from the center to the edges of the tissue.

When exploring how lateral inhibition is performed, we observe that the system is also bi-stable, but along a different diagonal axis: $\theta_3$ has low $D_{c1}$ and high $D_{c2}$ (low–high), while $\theta_4$ has the reverse (high–low). This corresponds precisely to the behavior of a lateral inhibition systems where any pair of neighboring cells should have opposite states for the patterning gene. Pattern formation in this case does not strictly depend on a dynamic propagation across the field.

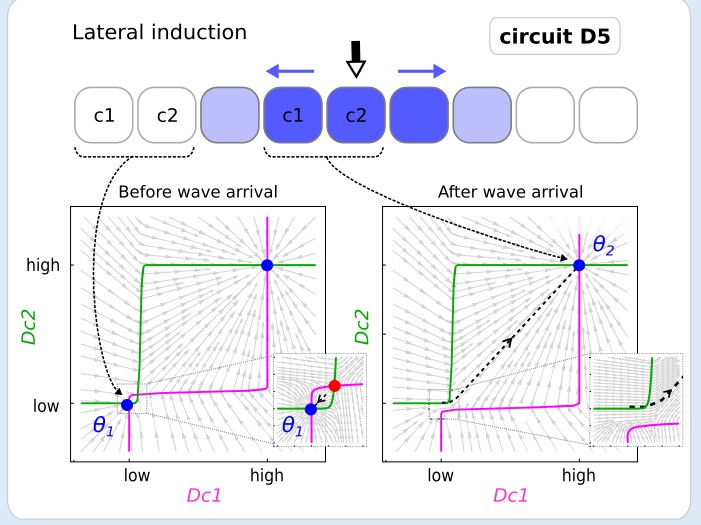

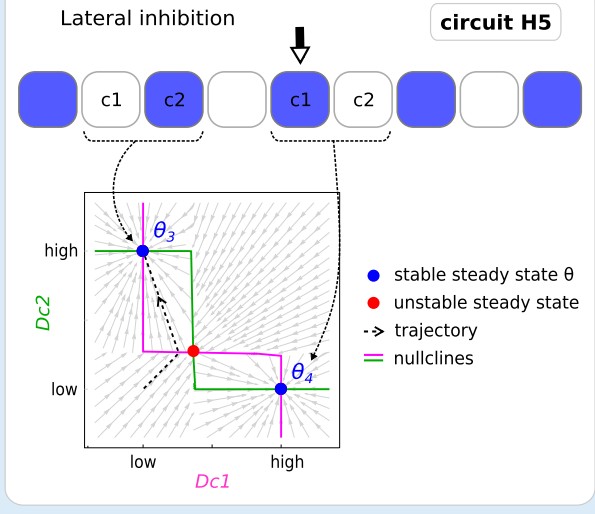

all minimal induction modules (Box 2). The analysis of hybrid C as it achieves inhibition (tissue A, $C = 0$) reveals a similar story—the arrangement of steady states is identical to that of H1 and characteristic of all inhibition modules (Box 2): two attractors at high–low ($\theta_4$) and low–high ($\theta_3$) with an unstable steady state in between. Thus, in terms of the distribution and movements of attractors, the dynamical mechanisms for the two functions of hybrid C are indeed equivalent to the dynamics of the underlying mono-functional

circuits. But how are the nullclines of the single bi-functional circuit able to specify the two very different arrangements of steady states?

In both D3 and H1, the nullclines for $D_{c1}$ and $D_{c2}$ (Fig 6B) take the form of sharp S-shaped sigmoids (which is a direct result of the non-linear gene regulation function chosen, see Box 1). However, their relative orientation is exactly opposite in D3 compared to H1 (i.e., the green nullcline for $D_{c2}$ is oriented like "S" in D3, but like "Z" in H1, and *vice versa* for $D_{c1}$). As we now focus on the more

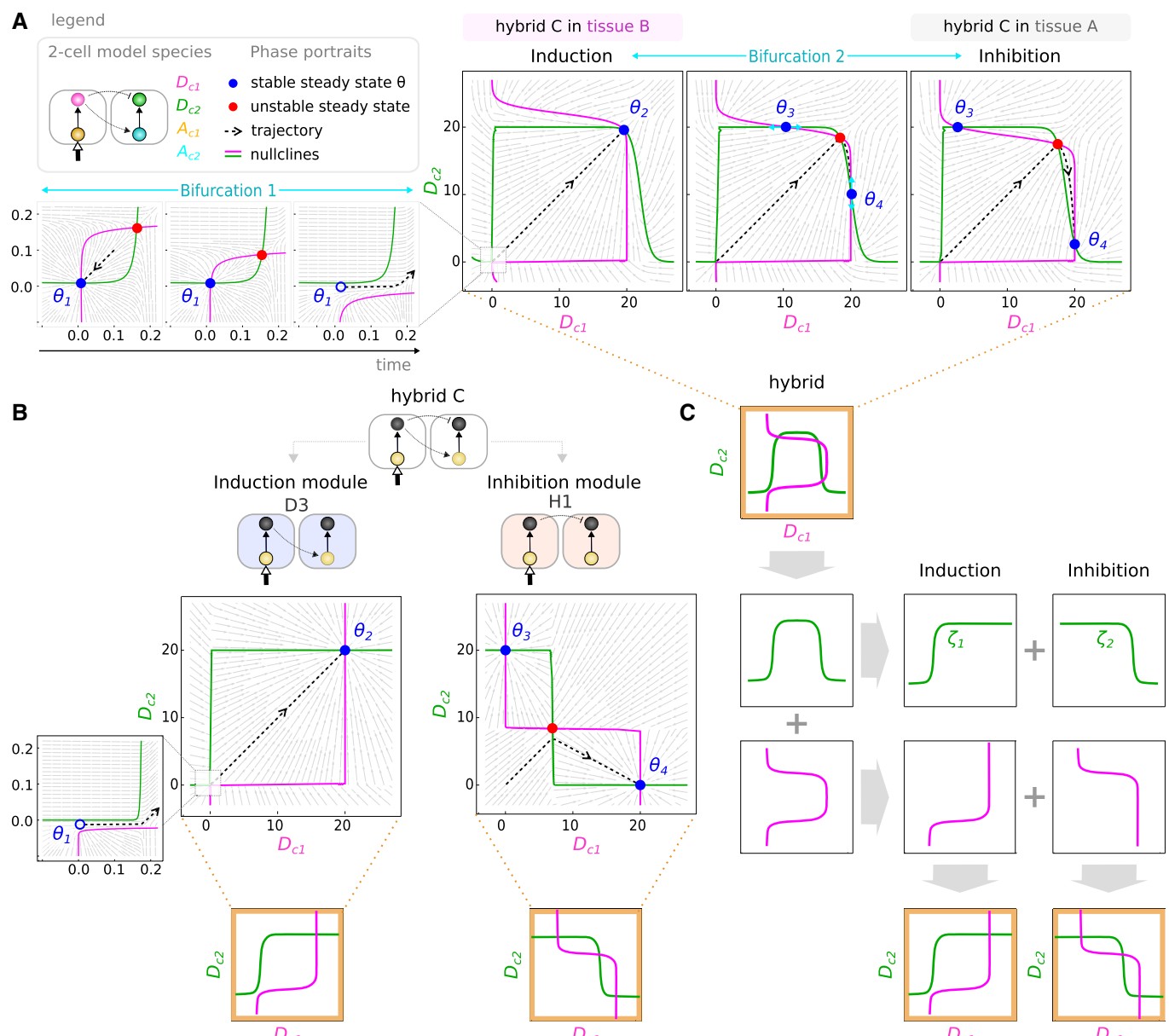

**Figure 6.    Function-switching mechanism and decomposability of hybrid circuits.**

A    How is hybrid C capable of performing both functions upon a change in the context signal? We use the simplified 2-cell model of Box 2 with parameters ($w_A = 0.41$, $w_B = 5.49$, $w_C = -0.30$, $\alpha_A = 6.93$, $\alpha_D = 12.79$). The context signal changes the position and number of steady states through a *pitchfork bifurcation* (Strogatz, 2014) (Bifurcation 2). This bifurcation drives the trajectory to access different attractors found in regions of the phase portrait corresponding to induction ($\theta_2$) and inhibition ($\theta_4$) patterns, respectively.

B    Phase portraits of the mono-functional modules (induction and inhibition) that build hybrid C.

C    The nullclines of hybrid C can be decomposed into sub-parts which correspond to the induction and inhibition modules.

complex nullclines of hybrid C, we observed they can indeed be decomposed into the two sigmoids of D3 and H1 (Fig 6C). For example, the green composite nullcline for $D_{c2}$ can be characterized as a $\cap$-shaped function. The "up" and "down" parts of the $\cap$-shape are constructed from the two sigmoids ($\zeta_1$ and $\zeta_2$) going in opposite directions—one responsible for the lateral induction function, and the other for lateral inhibition. Indeed only half of each $\cap$-shaped nullcline is required for each function. In the case of lateral induction the first sigmoid ($\zeta_1$) is necessary to produce the correct bi-stable attractors $\theta_1$ and $\theta_2$. The second sigmoid ($\zeta_2$), which brings the function back down to $D_{c2} = 0$, is present in the phase portrait but does not contribute at all to the lateral induction function. However, when the tissue context changes and the nullclines shift to alter the phase portrait, it is now the second sigmoid ($\zeta_2$) which is essential for the function (producing the $\theta_3$ and $\theta_4$ attractors). Again the first sigmoid is still present (on the left-hand side of the portrait), but is no longer important for the dynamics of the circuit.

We next explore how the hybrid is able to switch between functions just by changing the background expression level of one of the genes. How does a change on the external context input C affect the relative arrangement of the system's nullclines? Starting from the induction case, if the level of context signal C is reduced, the nullclines for $D_{c1}$ and $D_{c2}$ shift in such a way that the $\theta_2$ attractor bifurcates to create two new attractors ($\theta_3$ and $\theta_4$) with an unstable steady state in between (Bifurcation 2 in Fig 6A). As the value of C decreases further, the two new stable steady states move away from each other toward the high–low and low–high positions, thus re-creating the antagonistic bi-stable arrangement of the lateral inhibition function (a movie of this changing phase portrait shown in Appendix Fig S9 is provided as Movie EV1).

We have thus demonstrated in two distinct ways—both structurally and dynamically—that the hybrid circuit C is a simple composition of its underlying mono-functional "building blocks". The structural analysis (previous sections) showed that both sub-circuits (D3 and H1) exist within the design of the full circuit, and also that this relationship can be seen from the *stalactites* in the complexity atlas. In this section we have gone further, and shown that the same decomposability can be directly seen from the functional dynamics in the phase portraits.

### Dynamics of emergent bi-functional circuits

Can we now perform similar dynamical analysis to confirm that emergent circuits are different? The structural analysis has shown that in these cases the first function is indeed explained by a minimal sub-module within the circuit, but the second function is more complex, depends on all regulatory links of the full circuit, and therefore cannot be explained by a sub-module. Can we also show that the dynamics of this second more complex function is not equivalent to one of the minimal mono-functional building blocks?

We chose to study emergent $AI_1$ (Fig 7). We follow how the concentrations of the four species ($D_{c1}$, $A_{c1}$, $D_{c2}$, $A_{c2}$) evolve in time as the circuit performs inhibition (Fig 7A and B) or induction (Fig 7C and D). As predicted by the structural analysis, $AI_1$ achieves the first function using the same dynamics as its inhibition sub-module (H1). Indeed, the collection of steady states and positions of nullclines correspond to the characteristic arrangement of minimal inhibition circuits (Fig 7A, Box 2, Appendix Fig S8). However, in

order to achieve the second function, the circuit shows a phase portrait with no resemblance to those of minimal induction modules. The mono-functional circuits employ attractor-switching (orange boxes in Fig 7D), while $AI_1$ instead involves a dynamic known as *pursuit* (Verd *et al*, 2014, 2017). This type of mechanism has also been identified to drive pattern formation in real biological cases, for example, leading the transient domain shifts in the gap genes of *Drosophila* (Manu *et al*, 2009a,b).

During *pursuit*, the trajectory of a system follows a moving attractor which is finally reached. To visualize how $AI_1$ implements *pursuit*, we calculate instantaneous phase portraits at a series of time-points (Fig 7D). The system is permanently attracted toward the $\theta_5$ moving attractor. The change in position of this particular attractor is caused by two consecutive shifts in the position of the $D_{c1}$ (horizontal shift) and $D_{c2}$ (vertical shift) nullclines. As the $\theta_5$ attractor shifts its position faster than the current state of the system can manage, the trajectory is deviated and follows the moving attractor first horizontally (from $t_1$ to $t_3$ as $D_{c1}$ increases) then vertically (from $t_3$ to $t_4$ as $D_{c2}$ increases). Finally, the moving attractor is reached as it stands at a [high–high] induction state. Since only two of the four variables are plotted in these phase portraits, the attractor $\theta_5$ is probably in reality a manifold. However, this does not change the key conclusion that pursuit dynamics are involved rather than the bifurcation and attractor selection seen in D3.

Thus, we have been able to confirm that, while the lateral inhibition function is explained by the basic mono-functional circuit H1, the lateral induction function of this emergent circuit is qualitatively distinct from any of our minimal lateral induction circuits (D0–D5). Precisely, the induction behavior corresponds to a *pursuit* mechanism, which only emerges when the full bi-functional circuit is present.

### Gradual pattern transitions to mimic real biological systems

So far we have considered our model as representing how a single circuit could switch functions in two distinct tissues. As a final analysis, we here explored a slightly different scenario, in which both functions occur in the same tissue, but sequentially over time. This scenario is known in real biological systems: In the retina of *Drosophila* a wave of cellular differentiation known as the morphogenetic furrow progresses through the tissue and subsequently gives rise to a "salt and pepper" pattern of photoreceptor cells (Sato *et al*, 2013; Fig 8A). Likewise, in the chick inner ear a continuous domain of precursor cells known as patches of pro-sensory cells are induced by lateral induction, and this tissue subsequently creates a fine-grained pattern of neurogenic versus non-neurogenic (Daudet & Lewis, 2005; Petrovic *et al*, 2014). These cases bring up the interesting question of how one behavior transitions to the other. Unlike the cases discussed above, in which the two different tissue types are represented with a binary difference (C = 0 versus C = 1), in this case where a single tissue changes its behavior over time, the underlying context must change smoothly. So would the change in patterning mode also change smoothly, or instead exhibit an abrupt change?

We modeled time-dependent cues by gradually increasing the value of the context signal C from 0 to 1. A simulation of the hybrid circuit G' (Fig 8B) shows how the pattern develops over time. Interestingly, it shows how despite the smooth change in C, the switch in behavior from lateral induction behavior to lateral inhibition is sudden. The calculated phase portraits for this time-course explains

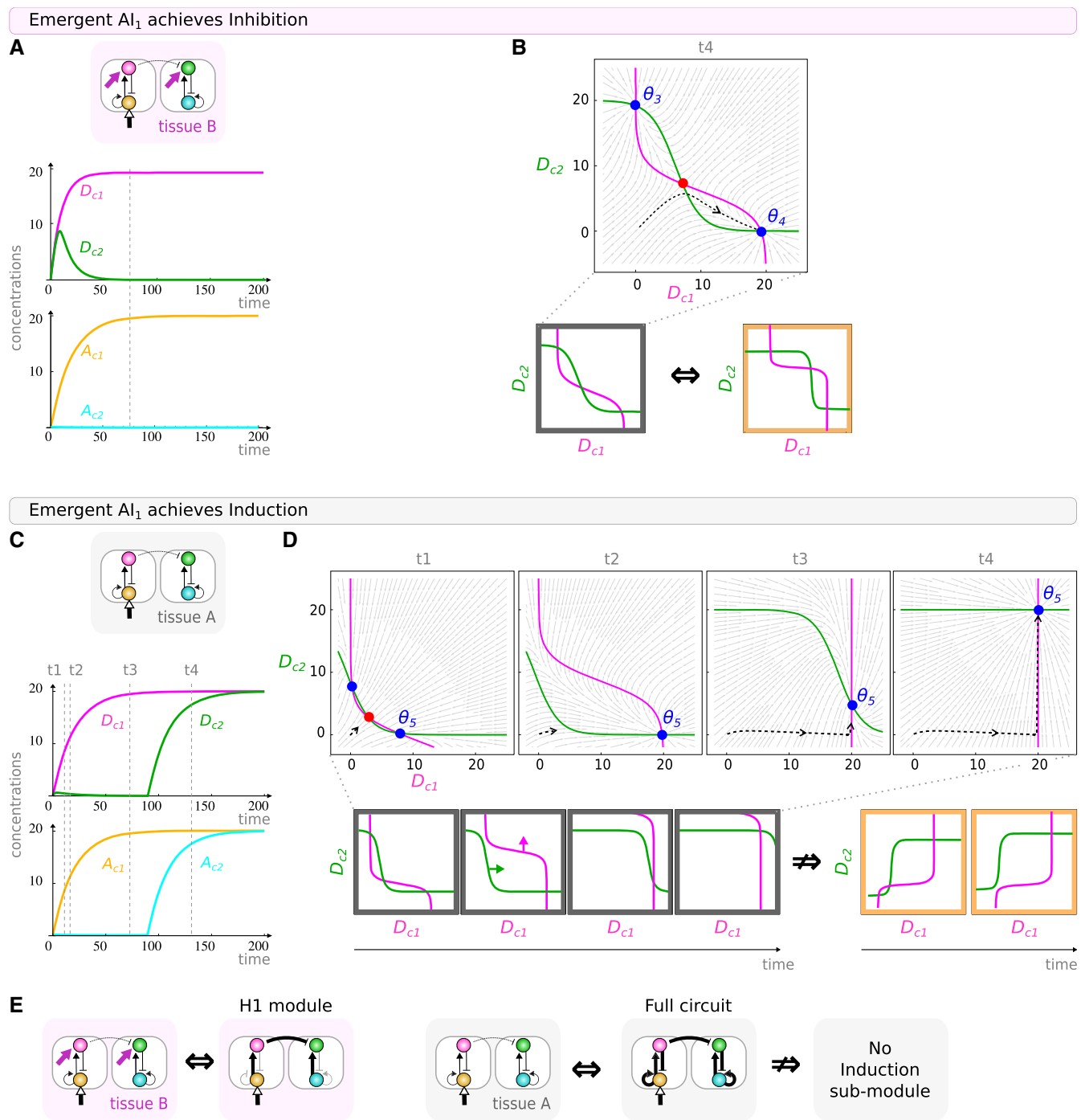

**Figure 7.  Dynamics of emergent circuits.**

A–D    For $AI_1$ circuit with parameters ($w_A = -0.05$, $w_B = -7.98$, $w_C = 6.47$, $w_D = 9.61$, $\alpha_A = 6.40$, $\alpha_D = 6.81$), we follow how concentrations of the four species ($D_{c1}$, $A_{c1}$, $D_{c2}$, $A_{c2}$) evolve in time as emergent circuit *Activation-Inhibition* $AI_1$ achieves (A, B) inhibition or (C, D) induction. In (B), we see how the phase portrait of $AI_1$ (gray box) is equivalent to that of the mono-functional inhibition circuits (orange box). (D) The lateral induction pattern results from a *pursuit* behavior (Verd *et al*, 2014, 2017) where the horizontal then vertical movement of the attractor $\theta_5$ deviates the trajectory which exhibits a sudden change in direction. This dynamic (gray boxes) is not equivalent to the dynamics seen in the minimal lateral induction circuits (orange boxes).

E    Structural view: The lateral inhibition function can be reduced to a sub-circuit which is indeed the minimal circuit H1, but the lateral induction function cannot—it requires the full circuit.

this abrupt transition: the high–high attractor, which provides the initial lateral induction behavior (equivalent to $\theta_2$ from Fig 6), is annihilated as the nullclines shift and two unstable steady states collide with it, leaving only the low–high and high–low attractors to provide the lateral inhibition behavior (known as a *subcritical pitchfork bifurcation*).

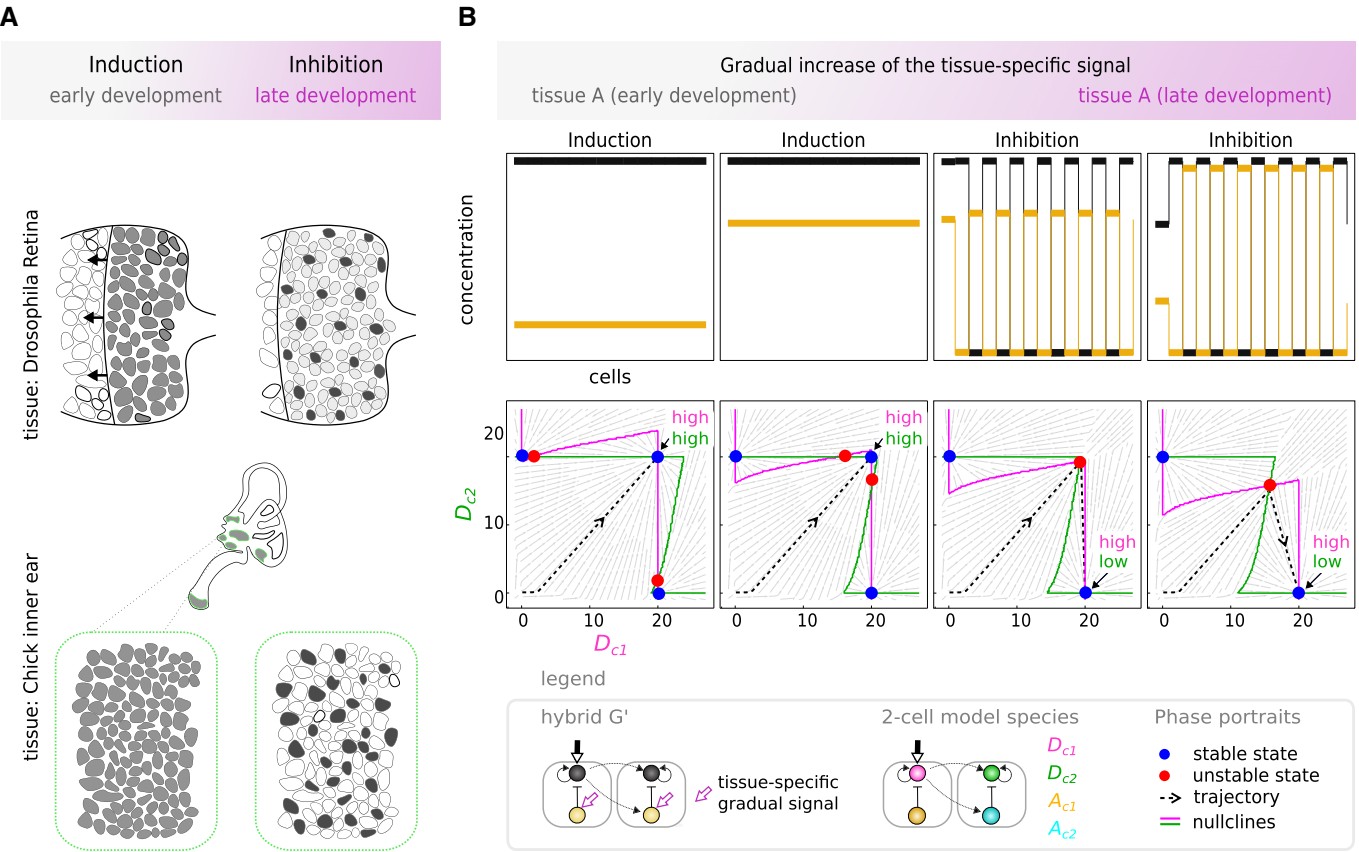

**Figure 8. Pattern transitions to model real biological systems.**

A Real biological systems are found where lateral induction precedes lateral inhibition in the same tissue. In the *Drosophila* eye, an initial wave of differentiation (morphogenetic furrow) progresses through the tissue to later give rise to a fine-grained pattern of R8 photoreceptor cells (Sato *et al*, 2013). In the chick's inner ear, a continuous domain of precursor cells, that is, patch of prosensory cells, gives rise to a mosaic of hair cells and supporting cells (Daudet & Lewis, 2005; Petrovic *et al*, 2014).

B We chose to model hybrid G' circuit with parameters ($w_A = -4.93$, $w_B = 3.22$, $w_C = 0.43$, $w_D = 0.14$, $\alpha_A = 15.86$, $\alpha_D = 8.62$). The context signal is treated as a time-dependent cue to model the transient nature of tissue environment. We show the final pattern and phase portraits for different values of $C$ from 0 to 1 (Box 2). The context $C$ behaves as a bifurcation parameter (*subcritical pitchfork* type): Two unstable states coalesce into a new fixed point that changes its stability: from stable to unstable. This type of bifurcations leads to a discontinuity in pattern transition.

Interestingly, the specific pattern transition shown in Fig 8B mimics the above mentioned real biological systems in which lateral induction precedes lateral inhibition. This sequence of events appears to be generally biologically relevant, as during organogenesis lateral induction is often an early event establishing a controlled uniform field of progenitor cells, which later undergo "salt and pepper" differentiation into two cell types. Finally, we should emphasize that the abrupt change in behavior shown for G' is not true for all hybrids. Hybrid C (shown in Appendix Fig S9) undergoes instead a smooth transition in which the amplitude of the "salt and pepper" pattern gradually decreases all the way to a very faint pattern, until finally becoming uniform.

## Discussion

Over the last couple of decades, scientists have built an encyclopedia of network motifs or building blocks able to explain the different

functions that real biological circuits perform (Milo *et al*, 2002; Shen-Orr *et al*, 2002; Alon, 2007; Davidson, 2010). This encyclopedia associates a given circuit (including parameter values) to a given function. Although this one-to-one relationship between structure and function has brought invaluable intuitive understanding on how biological processes are encoded, other features of real biological circuits are likely to complicate this simple picture. For example, genetic pleiotropy during embryo development highlights that genes, pathways, and circuits are often involved in multiple biological functions (Pires-daSilva & Sommer, 2003; Carroll *et al*, 2013). How is the structure of a circuit influenced if it performs multiple distinct functions in the same organism?

Our approach for studying multi-functional circuits had the following advantages. First, we imposed strong constraints on our model to seek the most highly compact multi-functional circuits. Our choice to seek the minimal collection of interacting genes able to achieve two qualitatively distinct functions allowed us to find circuits in which all components are essential to both functions

(Fig 1D). Importantly, these results go a step further than previous studies where the overlap between functional modules is only partial (Fig 1A–C). Second, in contrast to previous studies which focused on particular gene circuit architectures able to perform multiple functions [where these functions range from distinct dynamical behaviors (Panovska-Griffiths *et al*, 2013), to different single cell fates (Martin & Wagner, 2008) or distinct patterning functions (Palau-Ortin *et al*, 2015)], here our exhaustive theoretical search of circuit space was less biased and thus leads to more universal conclusions.

Through these approaches our key finding was revealed. Some bi-functional circuits can be seen as relatively intuitive composites of two simpler mono-functional modules. This can be true even for cases where the mono-functional "building blocks" are very tightly overlapping, that is, with no discernable *structural modularity*. In these *hybrid* cases, decomposability can be observed both at the level of circuit structure, and also in the dynamical mechanisms of the circuits (as seen in the analysis of phase portrait structure). However, we also found another class of bi-functional circuits— *emergent* circuits—which show no clear decomposability into pairs of simpler building blocks. These circuits are less easy to understand—both regarding this lack of modular decomposability, and also at the level of dynamics (Fig 7)—and so raise important questions about how we study the structure-function relationship for real circuits.

Intuition tends to guide many studies of real biological systems, and modularity is a well-documented finding in gene circuits, but there are clear dangers of trusting our intuition too much. The emergent circuits revealed here are not easy to understand, but they are just as robust as their hybrid counterparts (Appendix Fig S5C), and therefore equally biologically plausible. Data about the structure of real circuits tends to be incomplete and sometimes contain false positives (falsely inferred regulatory links). The structural differences between a hybrid circuit and an emergent one can be small (e.g., compare hybrid C with emergent $AI_1$) and so it may be difficult to rigorously distinguish which dynamical mechanism is more likely to explain a given set of functions. Theoretical mapping of complete atlases of possible circuits should therefore become invaluable guides to help our interpretation of empirical data and reduce the risks of jumping to false conclusions.

Although we explored only one particular pair of functions here (lateral induction and lateral inhibition), we expect that our primary conclusion will be a general finding. For many types of multi-functional circuits, a spectrum will exist, from those which can be intuitively decomposed into distinct sub-circuits underlying each function (*hybrid* class), to those in which the functions are more subtly distributed across a complex network (*emergent* class).

In addition to this direct consequence for understanding real gene circuits, our demonstration of strongly multi-functional circuits may have implications for some more general questions in biology. First, is the question of how densely biological information is encoded in the genome. A much discussed observation is that the number of genes an organism has does not correlate well with its organismal complexity (measured, e.g., as the number of distinct cell types in the organism). One broad approach to resolve this so called "G-value paradox" (Hahn & Wray, 2002), has been to search for further sources of complexity within the genome that could account for the acquisition of additional biological functions (Mattick *et al*, 2010; Schad *et al*, 2011)—such as modulation of chromatin (Kouzarides, 2007; Cairns, 2009), alternative splicing (Kim *et al*, 2007), the multi-functionality of proteins (Jeffery, 1999) or the newly discovered regulatory functions of dozens of types of non-protein coding RNAs such as lncRNAs or miRNAs (Sempere *et al*, 2006; Taft *et al*, 2007). However, knowing that multiple biological functions could be encoded in the same circuit—the same collection of genes—allows us to shift attention away from the idea that additional organismal complexity comes from additional molecular components. Instead, additional organismal complexity may derive from subtle adjustments to gene circuits which allow them to acquire extra functions without losing their original ones and without the need for additional molecular components—thus decoupling biological complexity from genomic complexity.

Second, the existence of highly compact multi-functional circuits may have consequences for evolution. *Structural modularity* (as seen in Fig 1A) has been proposed to confer advantages such as high robustness and evolvability: The failure of one module/function does not lead to the malfunctioning of the rest, and each module/function can evolve relatively independently (Raff & Conway Morris, 1996; Wagner & Altenberg, 1996; Kirschner & Gerhart, 1998; Brandon, 1999; von Dassow & Munro, 1999; Raff & Sly, 2000; Schlosser & Wagner, 2004). This could indeed be an *a priori* reason to expect modularity in circuits. However, the current paucity of data on the structures and dynamics of real circuits does not yet allow universal conclusions to be made, and so the reverse argument should still be explored: In some cases multi-functional (pleiotropic) gene circuits may lack strong modularity and therefore have an impact on evolution—solidifying a circuit into a given configuration with low evolvability, because most mutations would interfere with too many processes simultaneously. This scenario could be a plausible explanation for phenomena like the pentadactyl vertebrate limb. During evolution from fish to tetrapods, the geological record shows that the evolutionary variability in digit number has consistently decreased. Many phenotypic features of the mammalian limb have evolved dramatically (between, e.g., bats, whales, dogs, and humans), but the number of digits has remained fixed. This could potentially be explained by highly multi-functional developmental circuits, as the genes and signaling pathways important to this organ are also known to be essential for many other organs (eg. the tail bud, Sheeba *et al*, 2016). Indeed, this concept could go further—while structural modularity could explain the ability of two traits to evolve autonomously, the existence of compact overlapping modules could instead account for their covariation.

Last, we believe that the approach of mapping out landscapes of dynamical mechanisms, using tools such as the complexity atlas, has been and will remain important to engineering attempts to design and build new circuits synthetically (Matsuda *et al*, 2012; Schaerli *et al*, 2014; Matsuda *et al*, 2015). In this context, the finding of bi-functional motifs is of particular interest to synthetic biology. Already, distinct circuits have been successfully engineered that propagate a signal throughout a cell population under lateral induction (Matsuda *et al*, 2012) or lateral inhibition (Matsuda *et al*, 2015) modes—precisely the core circuits D2 and H2 have, respectively, been implemented. Despite the many technical challenges, the simple multi-functional designs found here could potentially be built synthetically as pattern-switch circuits able to achieve distinct patterns upon a tunable external input.

## Materials and Methods

### Exploring gene circuit space

A gene circuit consists of a topology, that defines the interactions among genes, and a specific set of parameters. A topology is composed of an inter-cellular and an intra-cellular circuit represented in the form of matrices. The inter-cellular matrix $W_{\text{inter}}$ describes how the signaling gene $D$ regulates genes in the neighboring cells, while the intra-cellular matrix $W_{\text{intra}}$ describes how genes interact among themselves within a cell. Within those matrices, positive values represent activation, negative values represent repression, and zero indicates no interaction. A parameter set consists of nine parameters: six regulatory parameters for the strengths of gene interactions, two parameters $\alpha$ (one per gene) to control gene-specific regulatory behavior, and one parameter $\beta$ to control the steepness of the regulatory function. To explore gene circuit space for solutions we first enumerate every possible topology (1,200 in total), then sample $10^7$ parameter sets per topology. The number of topologies can be calculated taking into account both the number of entries to $W_{\text{inter}}$ and $W_{\text{intra}}$ matrices—six entries with three choices each $(-1,0,1)$—and the choice of which of the genes is chosen to be the signaling gene $D$. From this potential number of topologies, isometric ones are removed. Parameters are chosen randomly within the following ranges: regulation $[-10;10]$, $\alpha$ $[-60;60]$, $\beta\{5, 10\}$.

### Defining functional gene circuits

Gene expression profile candidates to be classified as induction or inhibition must be stable and robust to developmental noise. That is, a gene profile is considered to have reached equilibrium when it remains stable for more than 100 consecutive time steps. Furthermore, equilibrium needs to be reached for four independent noise runs. First, in order to classify a pattern as induction, we measure the *expansion* level of a candidate gene at multiple time-points. The *expansion* of a gene is measured as the number of consecutive cells adjacent to the central cell for which its concentration is high, that is, above a certain threshold. This measure must increase at least five times during the simulation to finally be equal to the total number of cells in the tissue, that is, at equilibrium all cells express an homogeneous high expression level for the gene. Second, a gene was considered to have an inhibition pattern if its steady state expression level alternates between high and low expression states at least 13 times.

**Expanded View** for this article is available online.

## Acknowledgements

This work was supported by the Spanish Ministry of Economy, Industry and Competitiveness (MINECO), BFU2010-16428, the European Union's Horizon 2020 research and innovation program under Grant Agreement No. 670555, and the European Union Seventh Framework Program (FP7/2007-2013) under grant agreement 601062. We further acknowledge support of the Spanish Ministry of Economy, Industry and Competitiveness "Centro de Excelencia Severo Ochoa 2013–2017", SEV-2012-0208 and of the Cerca Programme/Generalitat de Catalunya.

## Author contributions

JS and AM conceived the project. JS, JC, and AJ wrote the manuscript. AJ performed the analysis.

## Conflict of interest

The authors declare that they have no conflict of interest.

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
