## [Review Process File · Molecular Systems Biology]

A spectrum of modularity in multi-functional gene circuits

Alba Jiménez, James Cotterell, Andreea Munteanu, James Sharpe

Corresponding author: James Sharpe, EMBL-CRG Systems Biology Research Unit

Review timeline:

Submission date:	27 September 2016
Editorial Decision:	18 November 2016
Revision received:	01 February 2017
Editorial Decision:	21 February 2017
Revision received:	16 March 2017
Accepted:	20 March 2017

Editor: Maria Polychronidou

Transaction Report:

1st Editorial Decision

18 November 2016

Thank you again for submitting your work to Molecular Systems Biology. Unfortunately, after a series of reminders, we have not managed to obtain a report from reviewer #2. In the interest of time, we have made a decision based on the other two reports. As you will see below, the reviewers raise a number of concerns, which we would ask you to address in a revision.

Without repeating all the points listed below, the reviewers think that the work needs to be better placed in the context of the existing literature i.e. by citing and discussing relevant previous work. Moreover, the part of the paper discussing the emergent networks and their biological implications (which as the reviewers point out, represents the main novel aspect of the study) should be explained and discussed more clearly. Considering the theoretical nature of the study, we would ask you to make sure that the main conclusions are easily accessible to the broad readership of Molecular Systems Biology (ideally by having the manuscript read by non-specialists). Moreover, it might be helpful to include didactic Boxes. Examples of didactic Boxes can be found at: msb.embopress.org/content/11/2/792 and <http://msb.embopress.org/content/11/7/818>).

REFeree REPORTS

Reviewer #1:

Following thematically upon some earlier work by some of the same authors (Cotterell & Sharpe, MSB, 2010), Jimenez et al. build a complexity atlas of gene regulatory circuits that can perform either lateral induction or lateral inhibition --- two important developmental functions of regulatory circuits. They use this atlas to identify minimal circuit topologies that have

one of these two functions, and show that these functions can be performed with a variety of dynamical mechanisms. This finding complements the author's earlier work on stripe-forming networks (Schaerli et al., Nat. Comm., 2014; Jimenez et al., PNAS, 2015). The authors then move on to identify circuit topologies that can perform both functions. They find that bi-functionality can be achieved in (1) hybrid circuits, which combine topological elements of a minimal circuit that can perform one function with topological elements of a minimal circuit that can perform the other function, and (2) emergent circuits, which are complex in structure and do not achieve bi-functionality by combining elements of mono-functional circuits. They then explore the dynamical mechanisms by which patterning is achieved and find that the dynamics of hybrid circuits can be decomposed into the dynamics of their constituent circuits, just as their topologies can. For emergent bi-functional circuits, the dynamics are not decomposable, pointing to limits to the modularity of multifunctional gene circuits. Finally, the authors study how both induction and inhibition can occur in the same tissue, but sequentially over time.

I think this paper is timely and important and goes well beyond the author's previous work, even though it is clearly related. It contains significant conceptual advances that build upon a growing literature on the relationship between form and function in gene regulatory circuits. This paper is likely to be read by a broad audience, including systems biologists, developmental biologists, and synthetic biologists, as well as computer scientists who study the kinds of models investigated here.

Below, I have listed a few comments that I hope will help the authors to improve an already very good manuscript:

Major comments:

There are some highly relevant papers on the relationship between form and function in gene regulatory circuits that the authors have not cited. I think the manuscript would benefit from the inclusion and discussion of the following references:

Ingram, Stumpf, & Stark, 2006, BMC Genomics, 7:108.
 Macia, Widder, and Sole, 2009, BMC Syst Biol., 3:84.
 Payne & Wagner, 2015, Sci Rep, 5:13015.
 Sorrells, Booth, Tuch, & Johnson, 2015, Nature, 523:361.
 Ahnert & Fink, 2016, J. Roy. Soc. Interface, 13:20160179.

For example, Payne & Wagner (2015) showed that multifunctionality constrains circuit architecture (i.e., there are fewer circuit topologies that are multifunctional than monofunctional), a finding that is highly relevant to the work presented here. This theoretical finding is supported empirically by the work of Sorrells et al., who study an example of what Jimenez et al. in Fig. 1 call "partial module overlap".

I'm having difficulty with your classification of regulatory logic according to the parameter α . In particular, I find your definition of an "AND" function troubling ($\alpha > 15$), because to me, an "AND" function implies that two distinct regulatory inputs need to be present and active. This does not appear to be the case in your model, as far as I can tell. For example, if gene A is only regulated by gene D, and gene A has $\alpha = 16$, then gene A will be classified as implementing "AND" logic. This is strange because gene A only has one input in this example. Moreover, gene A could be activated if the regulatory weight w_2 is sufficiently high. Could you please explain how you justify this definition of an "AND" function? Additionally, please explain why you choose 15 as the threshold to define an "AND" function?

It would be useful to know which of the two mono-functions each bi-functional topology is biased toward. For each bifunctional topology (green nodes in Fig. 3), please show the proportion of the 10^7 sampled parameters that yield induction and the proportion that yield inhibition. This could be done by showing each node as a pie chart, for example.

Minor comments:

Abstract. The terms "hybrid circuits" and "emergent circuits" need to be defined upon their first use in the abstract.

Fig. 2C, right box named "parameter-set". Give each matrix entry its own name, so you can reference these later in the text. I.e., the second entry in the inter-cellular circuit should be labeled w_2 instead of zero. Label the entries in the intra-cellular matrix accordingly.

In the "finding minimal circuits" section, please explain how there are 1200 topologies. Since there are in total only 6 entries to the W_{intra} and W_{inter} matrices, and each entry can take on one of three values (-1,0,1), then there should be $3^6 = 729$ topologies, not 1200. I'm sure I'm missing something here, but I'm also sure that I won't be alone in missing it.

In the "finding minimal circuits" section, please cite Schuster et al. 1994 Proc Roy Soc B 255:279-284 after you mention "neutral networks."

In the "dynamics and decomposability" section, give a back-reference to Fig. 5 after "and we chose the circuit hybrid C..." This will remind the reader where that circuit resides in the atlas.

Fig. 6. It is confusing that you use α to represent attractors, when you previously used it to define regulatory logic (Supp Fig. 1). Please use different symbols for these two cases.

Supplementary Fig. 1. In panel A, the sigmoidal function should make it clear that "input" can actually represent the sum of expression levels of more than one gene. For panels C-D, the caption should describe what the data points, blue lines, and red lines mean.

Supplementary Fig. 3. In the caption, w_A and w_B are used to define the strength of intra- and inter-cellular interactions, respectively. This is confusing because in Fig. 2C, these interactions are represented by 2x2 and 2x1 matrices. I suppose this choice was based on the fact that there is only one intra- and one inter-cellular interaction in the minimal circuits H0 through H5 that are shown in Fig. S3B. Nonetheless, this could be made more clear by labeling the edges in Fig. S3B, or using the same entry labels as in Fig. 2C, following my earlier comment.

Trivia:

Abstract:

"A central question in systems biology is to understand the relationship between a circuit's structure and its function..." It's clear what you mean here, but this is not a question. Please rephrase.

Typo: "...structural modularity ? they can switch..."

Introduction. The authors state that "In this study we address both questions", but the preceding paragraph listed 3 questions.

Material and Methods.

"among them within" -> "among themselves within", "to control for the" -> "to control the", "34.155.071" -> "34,155,071", "9.732.253" -> "9,732,253", "paremeter" -> "parameter", "takes the form of a sigmoid function ()" -> "takes the form of a sigmoid function." Please explain in words what "one Hamming Distance apart" means. In your case, it's just that two circuits differ in a single regulatory interaction. I think it would be easier for the reader if you just said that.

Supplementary Figures:

Fig. 1: "...absence of input of despite..." -> "absence of input or despite..."

Fig. 2: "beneficial role into refining" -> "beneficial role in refining"

References:

Some journal names are abbreviated, others are not. Some article titles have the first letter of every word in caps, and others don't. Please follow the journal guidelines and be consistent throughout.

There is a mistake in the 4th author's name in the Palmeirim et al. reference.

Throughout the manuscript:

As a matter of style, only variable names should be italicized. So, for example, use $\$W_{\{\mathrm{intra}\}}\$$ rather than $\$W_{\{intra\}}\$$. This will change your life.

In equations, use \backslashexp so that the exponential function is not italicized.

Reviewer #3:

This paper presents results on the networks able to perform dual functionalities relevant for biology, essentially alternating or propagating pattern. The authors enumerate networks for both function, taken independently or simultaneously (via introduction of an external control), and describe two types of networks, some being module, others being emergent, the main result of the paper being that there are "limits to the modularity of multi functional gene circuits".

There are two main aspects of the paper that I wish to comment separately. The first aspect is on the general level of the "modularity vs emergent" behaviour, which is the main focus (and title) of the paper. I find that most of the discussion relies (too much) on the prejudice that networks are expected to be modular (as often claimed). However 1> this is in my opinion a prejudice 2> alternative explanations have been addressed in other papers in better ways. I think some more credit and reference to previous literature should be added. For instance, the authors cite Kashtan et al, 2005 in Figure 1 caption, but this paper actually clearly discusses the fact that modularity is not really expected a priori, and indeed exhibit examples of optimized networks where there is no modularity, e.g. Fig 2a of that paper, which would be qualified as "emergent" by the authors. The explanation offered in general is that modularity emerges from other evolutionary constraints (in Kashtan: alternating evolutionary pressure). Discussions on those aspects are missing, despite the fact that they have motivated other kinds of approaches such as in silico evolution. Emergence is also something very well known in neuroscience, e.g. Sussilo and Abbott, 2009 show how we can take a complex interaction networks and essentially extract any functional behaviour. So I do not find at all surprising that "emergent" networks exist, this is very much expected. This paper only gives a potentially new example of this, it does not shed any particular new light on this specific aspect and I find the general conclusions drawn overstretched and not particularly original given this only example.

The second aspect, the description of networks able to have two different functions in this patterning context brings some perspective to this problem, by enumerating all possible small networks/associated parameters, and at least finding a non-trivial emergent networks. I also found the biological implications/examples potentially interesting, although of course at this stage this is nothing more than a theoretical proposal. I nevertheless have a couple of comments to make this part of the paper more understandable, because as is I find that results on the modularity are kind of expected, while the part on the emergent dynamics is relatively hard to grasp. Finally, I think some of the results are relatively close to previous work of Corson and Siggia, 2012, performed on a more realistic biological context and some comparison would be welcome.

Some comments:

- the authors often cite recent works on multi-functional networks being bistable/oscillating. There is actually older literature from the 2000s on this, for instance the Mixed Feedback Loop module has been shown to realize this with simple changes of parameters (Francois and Hakim, 2005). There also is a paper by Rouault and Hakim, Biophys J 2011 that is very similar in spirit to what is described here (evolution of lateral patterning) that could be cited in my opinion.

- the « single function » networks are essentially trivial. These are cell to cell positive/negative feedback loops depending on either inDuction or inHibition. The only complexity is on the combinatorics: since there are two genes per cell, one can either activate an activator, repress a

repressor. Figure 6 on the modularity is a classical example of bifurcation to bistability with the external variable as an external parameter. I do not think we learn much there, and all these discussions could be shortened.

- the real novelty of the paper is the emergent network that can not be simply disentangled into functional module. In that regard, I find the explanation of Fig 7 particularly unclear. First, since there is a time component it would be very useful to actually show the dynamics of the 4 different genes with respect to time (and not only the bifurcation diagram) so that we can relate to the explanation on the right side of the figure. Those bifurcations diagram also are themselves quite confusing: for instance there is a new fixed point appearing out of nowhere that is not even plotted in the A/D plane and emerges out of the figure, so we have no real idea of what happens. The full null clines should be drawn so that we can understand the dynamics.

- as far as I can see, the behaviour of this emergent networks bears strong resemblance with a recent « geometric » analysis by Corson and Siggia, PNAS 2012, where they show using pure geometries in phase space how one can change direction of the trajectory of the system and get to a new basin of attraction on the *C elegans* vulva example. The problem is very close in spirit because changes of basins is due to cell-cell interaction as well, but Corson and Siggia are also predictive of actual data. This raises two questions: 1> how are the topology and geometry in phase space presented here comparable to this previous work ? and 2> does that mean that we should focus on phase space geometry rather than network topology, as proposed by Corson and Siggia ?

1st Revision - authors' response

01 February 2017

Text continues on next page.

Response to the comments of the referees

We thank the reviewers for their comments about our study and for suggesting ways of improvement. In response, we have revised both the manuscript and the appendix throughout their length, and the specific modifications suggested by the referees have all been taken into account. Particularly, in order to properly acknowledge previous studies related to our work, the introduction has been revised and includes a classification of distinct types of multi-functional circuits. Furthermore, for clarity purposes, the structure of the main text has been revised to include 2 new didactic Boxes that will help non-specialist readers follow the manuscript. Last, note that, following the general comments of reviewer #3, we have changed the title of our manuscript from "*Limits to the modularity of multi-functional gene circuits*" to "*A spectrum of modularity in multi-functional gene circuits*".

Reviewer #1:

Following thematically upon some earlier work by some of the same authors (Cotterell & Sharpe, MSB, 2010), Jimenez et al. build a complexity atlas of gene regulatory circuits that can perform either lateral induction or lateral inhibition --- two important developmental functions of regulatory circuits. They use this atlas to identify minimal circuit topologies that have one of these two functions, and show that these functions can be performed with a variety of dynamical mechanisms. This finding complements the author's earlier work on stripe-forming networks (Schaerli et al., Nat.Comm., 2014; Jimenez et al., PNAS, 2015). The authors then move on to identify circuit topologies that can perform both functions. They find that bi-functionality can be achieved in (1) hybrid circuits, which combine topological elements of a minimal circuit that can perform one function with topological elements of a minimal circuit that can perform the other function, and (2) emergent circuits, which are complex in structure and do not achieve bi-functionality by combining elements of mono-functional circuits. They then explore the dynamical mechanisms by which patterning is achieved and find that the dynamics of hybrid circuits can be decomposed into the dynamics of their constituent circuits, just as their topologies can. For emergent bi-functional circuits, the dynamics are not decomposable, pointing to limits to the modularity of multi-functional gene circuits. Finally, the authors study how both induction and inhibition can occur in the same tissue, but sequentially over time.

I think this paper is timely and important and goes well beyond the author's previous work, even though it is clearly related. It contains significant conceptual advances that build upon a growing literature on the relationship between form and function in gene regulatory circuits. This paper is likely to be read by a broad audience, including systems biologists, developmental biologists, and synthetic biologists, as well as computer scientists who study the kinds of models investigated here.

Below, I have listed a few comments that I hope will help the authors to improve an already very good manuscript:

Major comments:

There are some highly relevant papers on the relationship between form and function in gene regulatory circuits that the authors have not cited. I think the manuscript would benefit from the inclusion and discussion of the following references:

Ingram, Stumpf, & Stark, 2006, BMC Genomics, 7:108.

Macia, Widder, and Sole, 2009, BMC Syst Biol., 3:84.

Payne & Wagner, 2015, Sci Rep, 5:13015.

Sorrells, Booth, Tuch, & Johnson, 2015, Nature, 523:361.

Ahnert & Fink, 2016, J. Roy. Soc. Interface, 13:20160179.

We have now included in the introduction a classification of distinct types of multifunctionality and also made important changes to the text throughout. In order to build this classification, most of the references given by the referee have been key. Ingram et al. has been cited in the introduction as an excellent example of a decision-making function, Macia et al. has been cited in section "Strongly bi-functional circuits" in relation to the idea of *specialized* versus *flexible* circuits. Sorrells et al. has been cited in Figure 1 in relation to partial module overlap. Finally, we have cited Ahnert and Fink in the

introduction as a case study of the complex relationship between structure and function.

For example, Payne & Wagner (2015) showed that multi-functionality constrains circuit architecture (i.e., there are fewer circuit topologies that are multi-functional than mono-functional), a finding that is highly relevant to the work presented here. This theoretical finding is supported empirically by the work of Sorrells et al., who study an example of what Jimenez et al. in Fig. 1 call "partial module overlap".

We now cite Payne & Wagner (2015) previous conclusions on how multi-functionality constrains circuit architecture. On a new paragraph at the beginning of section "Strongly bi-functional circuits", we provide the ratios of mono-functional versus bi-functional topologies, which are indeed very similar to those of Payne & Wagner (2015).

I'm having difficulty with your classification of regulatory logic according to the parameter α . In particular, I find your definition of an "AND" function troubling ($\alpha > 15$), because to me, an "AND" function implies that two distinct regulatory inputs need to be present and active. This does not appear to be the case in your model, as far as I can tell. For example, if gene A is only regulated by gene D, and gene A has $\alpha = 16$, then gene A will be classified as implementing "AND" logic. This is strange because gene A only has one input in this example. Moreover, gene A could be activated if the regulatory weight w_2 is sufficiently high. Could you please explain how you justify this definition of an "AND" function? Additionally, please explain why you choose 15 as the threshold to define an "AND" function?

We agree with the referee and have decided to not use the AND/OR analogy. Indeed, as pointed out, the response to lower or high input do not necessarily correspond to the classic boolean AND/OR responses. Instead, in the new Box 1 we simply distinguish between two regulatory logics: constitutive and non-constitutive. In agreement to this, changes have also been made in Appendix Figure S3.

It would be useful to know which of the two mono-functions each bi-functional topology is biased toward. For each bi-functional topology (green nodes in Fig. 3), please show the proportion of the 10^7 sampled parameters that yield induction and the proportion that yield inhibition. This could be done by showing each node as a pie chart, for example.

Following the referee's remark, we created pie-charts for each bi-functional topology of Fig.3. For each topology, pie-charts show the proportion of sampled parameters that yield induction or inhibition. Since this is a very dense and large amount of information, we added a new Appendix Figure S4. Interestingly, we noticed that a majority of bi-functional topologies are strongly biased towards one of the functions, i.e they are *specialized* (Macia et al. 2009). We further comment on this observation in a new paragraph at the beginning of section "Strongly bi-functional circuits".

Minor comments:

Abstract. The terms "hybrid circuits" and "emergent circuits" need to be defined upon their first use in the abstract.

The abstract has been re-written to make a more explicit description of these two terms.

Fig. 2C, right box named "parameter-set". Give each matrix entry its own name, so you can reference these later in the text. i.e., the second entry in the inter-cellular circuit should be labeled w_2 instead of zero. Label the entries in the intra-cellular matrix accordingly.

Supplementary Fig. 3. In the caption, w_A and w_B are used to define the strength of intra- and inter-cellular interactions, respectively. This is confusing because in Fig. 2C, these interactions are represented by 2×2 and 2×1 matrices. I suppose this choice was based on the fact that there is only one intra- and one inter-cellular interaction in the minimal circuits H0 through H5 that are shown in Fig. S3B. Nonetheless, this could be made more clear by labeling the edges in Fig. S3B, or using the same entry labels as in Fig. 2C, following my earlier comment.

We strongly agree with the referee and have made the following changes. In Fig. 2B, we have given each

matrix its own name W_{inter} and W_{intra} as this is the first time these matrices are introduced. Also, we have labeled the entries in those matrices as suggested. Later in the text, Box 1 references these matrices to describe the gene regulatory model. This new labeling helps clarity throughout the text as the reader can always identify a regulatory link through the index w_n in a consistent manner. We thus use this notation for every parameter-set analyzed in Figures 6, 7 and 8. The same applies to the former Fig. S3B or new Appendix Figure S2.

In the "finding minimal circuits" section, please explain how there are 1200 topologies. Since there are in total only 6 entries to the W_{intra} and W_{inter} matrices, and each entry can take on one of three values (-1,0,1), then there should be $3^6 = 729$ topologies, not 1200. I'm sure I'm missing something here, but I'm also sure that I won't be alone in missing it.

Indeed, the exhaustive listing of possible topologies needs some clarification. As pointed out, the number of possible matrices without taking into account the trigger input is 729. However, for each of the 729 topologies, the trigger can be received by either gene. From this 729x2 possible topologies, we have leaved out the ones that are symmetrical or isometric. This is explained with further detail in the Methods, in "Exploring gene circuit space" section.

In the "finding minimal circuits" section, please cite Schuster et al. 1994 Proc Roy Soc B 255:279-284 after you mention "neutral networks." In the "dynamics and decomposability" section, give a back-reference to Fig. 5 after "and we chose the circuit hybrid C..." This will remind the reader where that circuit resides in the atlas.

Both comments have been taken into account.

Fig. 6. It is confusing that you use α to represent attractors, when you previously used it to define regulatory logic (Supp Fig. 1). Please use different symbols for these two cases.

We chose to keep α and β as parameters of the regulatory function. Instead, we now use θ to reference attractors.

Supplementary Fig. 1. In panel A, the sigmoidal function should make it clear that "input" can actually represent the sum of expression levels of more than one gene. For panels C-D, the caption should describe what the data points, blue lines, and red lines mean.

According to the new structure of the text that includes a didactic Box 1 to clarify the theoretical aspects of the manuscript, we chose to move the graph of input-to-output of the regulatory function (previously in former Supplementary Fig. 1.) to the new Box 1 in the main text. Particularly, we add a cartoon showing how the input in the x-axis corresponds to the sum of expression levels of more than one gene.

Trivial:

– Abstract: "A central question in systems biology is to understand the relationship between a circuit's structure and its function..." It's clear what you mean here, but this is not a question. Please rephrase.

– Typo: "...structural modularity ? they can switch..."

– Introduction. The authors state that "In this study we address both questions", but the preceding paragraph listed 3 questions.

– Material and Methods. "among them within" -> "among themselves within", "to control for the" -> "to control the", "34.155.071" -> "34,155,071", "9.732.253" -> "9,732,253", "paremeter" -> "parameter", "takes the form of a sigmoid function ()" -> "takes the form of a sigmoid function." Please explain in words what "one Hamming Distance apart" means. In your case, it's just that two circuits differ in a single regulatory interaction. I think it would be easier for the reader if you just said that.

– Supplementary Figures:

Fig. 1: "...absence of input of despite..." -> "absence of input or despite..."

Fig. 2: "beneficial role into refining" -> "beneficial role in refining"

– References:

Some journal names are abbreviated, others are not. Some article titles have the first letter of every word in

caps, and others don't. Please follow the journal guidelines and be consistent throughout. There is a mistake in the 4th author's name in the Palmeirim et al. reference.

– Throughout the manuscript:

As a matter of style, only variable names should be italicized. So, for example, use W_{intra} rather than W_{intra} . This will change your life. In equations, use \exp so that the exponential function is not italicized.

We thank the reviewer for being so exhaustive in picking up these mistakes – all of which have now been corrected.

Reviewer #3:

This paper presents results on the networks able to perform dual functionalities relevant for biology, essentially alternating or propagating pattern. The authors enumerate networks for both function, taken independently or simultaneously (via introduction of an external control), and describe two types of networks, some being module, others being emergent, the main result of the paper being that there are "limits to the modularity of multi-functional gene circuits".

There are two main aspects of the paper that I wish to comment separately. The first aspect is on the general level of the "modularity vs emergent" behaviour, which is the main focus (and title) of the paper. I find that most of the discussion relies (too much) on the prejudice that networks are expected to be modular (as often claimed). However 1> this is in my opinion a prejudice 2> alternative explanations have been addressed in other papers in better ways. I think some more credit and reference to previous literature should be added. For instance, the authors cite Kashtan et al, 2005 in Figure 1 caption, but this paper actually clearly discusses the fact that modularity is not really expected a priori, and indeed exhibit examples of optimized networks where there is no modularity, e.g. Fig 2a of that paper, which would be qualified as "emergent" by the authors. The explanation offered in general is that modularity emerges from other evolutionary constraints (in Kashtan: alternating evolutionary pressure). Discussions on those aspects are missing, despite the fact that they have motivated other kinds of approaches such as *in silico* evolution.

We agree that we may have got the balance wrong. Indeed we agree with the reviewer's own perspective, but nevertheless feel strongly that overall the literature is still biased towards modularity. Indeed, the reviewer mentions that "Kashtan et al, 2005 ... clearly discusses the fact that modularity is not really expected a priori". However, while they indeed express the questioning of why circuits should evolve modularity, they do not question the bias of opinion that real circuits are usually modular:

- 1) The abstract starts as follows: "Biological networks have an inherent simplicity: they are modular with a design that can be separated into units that perform almost independently. Furthermore, they show reuse of recurring patterns termed network motifs. Little is known about the evolutionary origin of these properties."
- 2) The Introduction starts as follows: "Biological and engineered systems share general design features: they display modularity, defined as the separability of the design into units that perform independently, at least to a first approximation (1–3, 5). Furthermore, they show reuse of certain circuit patterns, termed network motifs (6–11), in many different parts of the system. These features allow construction of extremely complex systems by using simple building blocks."

And numerous other papers start with similar statements such as Solé & Valverde, 2008; Clune & Lipson, 2013; Wagner, Mezey & Calabretta, 2001; Wagner, Pavlicev & Cheverud, 2007 and others.

The non-modularity shown in Kashtan Figure 2a is because the circuit shown is mono-functional. Indeed, in the modular circuits shown later, the two modules did not reflect the two alternative functions of the whole circuit (called G1 and G2 by the authors). Instead, the modules reflected two "subproblems" (X XOR Y) and (Z XOR W), each of which were necessary for *both* of the functions G1 and G2. We have therefore made clearer in the text that our study only concerns the question of modularity in multi-functional scenarios. There is no reason to expect modularity in mono-functional scenarios.

Emergence is also something very well known in neuroscience, e.g. Sussilo and Abbott, 2009 show how we can take a complex interaction networks and essentially extract any functional behaviour. So I do not find at all surprising that "emergent" networks exist, this is very much expected. This paper only gives a potentially new example of this, it does not shed any particular new light on this specific aspect and I find the general conclusions drawn overstretched and not particularly original given this only example.

Indeed emergence is a phenomenon known in a very wide variety of dynamical systems. In general it describes when a "higher-level" behavior is seen which is not easily explainable or reducible to the behavior of sub-parts of the system. Just finding another example of emergence *per se* might have

limited interest. However, our study is focused multi-functionality specifically (not just the structure-function relationship in general) and on defining different ways in which multi-functionality arises. We have now included a new paragraph in the introduction to highlight where the novelty comes in our own study:

- (a) a focus on decomposability – specifically which nodes and which links of multi-functional circuits are involved in each of its functions.
- (b) an attempt to understand the structure-function relationship in term of both decomposable structure and decomposable dynamics.
- (c) an attempt to go beyond the analysis of just one or two chosen circuits, and instead perform a systematic survey across a given class of circuits, so that more general conclusions can be drawn.

The second aspect, the description of networks able to have two different functions in this patterning context brings some perspective to this problem, by enumerating all possible small networks/associated parameters, and at least finding a non-trivial emergent networks. I also found the biological implications/examples potentially interesting, although of course at this stage this is nothing more than a theoretical proposal. I nevertheless have a couple of comments to make this part of the paper more understandable, because as I find that results on the modularity are kind of expected, while the part on the emergent dynamics is relatively hard to grasp. Finally, I think some of the results are relatively close to previous work of Corson and Siggia, 2012, performed on a more realistic biological context and some comparison would be welcome.

Some comments:

- the authors often cite recent works on multi-functional networks being bistable/oscillating. There is actually older literature from the 2000s on this, for instance the Mixed Feedback Loop module has been shown to realize this with simple changes of parameters (Francois and Hakim, 2005). There also is a paper by Rouault and Hakim, *Biophys J* 2012 that is very similar in spirit to what is described here (evolution of lateral patterning) that could be cited in my opinion.

We thank the reviewer for these two references which we have indeed missed, and are very interesting. Both Francois and Hakim, 2005 and Rouault and Hakim, 2012 have added to the introduction as examples of multi-functional circuits which transition between two qualitatively distinct behaviors (oscillatory behavior and bi-stability) upon a change in parameters. Furthermore, Rouault and Hakim 2012 has also been cited in section “The dynamics and decomposability of hybrid circuits” in relation to studies that performed geometric analysis of a circuit's phase space.

- the « single function » networks are essentially trivial. These are cell to cell positive/negative feedback loops depending on either inDuction or inHibition. The only complexity is on the combinatorics: since there are two genes per cell, one can either activate an activator, repress a repressor. Figure 6 on the modularity is a classical example of bifurcation to bistability with the external variable as an external parameter. I do not think we learn much there, and all these discussions could be shortened.

We believe that the description of mono-functional circuits is essential, as it forms the basis of the subsequent analysis of modularity in bi-functional circuits. Although the transition of hybrid C from induction to inhibition in Figure 6 is indeed a classic pitchfork bifurcation, we believe that these phase portraits (which represent the states of two distinct cells) go beyond the classical bifurcation diagrams where a given single cell can access alternative differentiation states (Huang et al (2007)) (see last paragraph of this letter). Indeed, in the traditional cell-type phase portrait, different attractors represent different cell types or states. By contrast, in this diagram the two different attractors do not represent differences in cell states, but rather differences in the spatial pattern of these 2 cell states. We have tried to make this point clearer in the text now, and have also included Box 2 to help shorten the discussion this section (indeed, this Box 2 helps us to first comment separately on the dynamics of induction and inhibition). We believe this helps fluidity in the text.

- the real novelty of the paper is the emergent network that can not be simply disentangled into functional module. In that regard, I find the explanation of Fig 7 particularly unclear. First, since there is a time component it would be very useful to actually show the dynamics of the 4 different genes with respect to time (and not only the bifurcation diagram) so that we can relate to the explanation on the right side of the figure. Those bifurcations diagram also are themselves quite confusing: for instance there is a new fixed

point appearing out of nowhere that is not even plotted in the A/D plane and emerges out of the figure, so we have no real idea of what happens. The full nullclines should be drawn so that we can understand the dynamics.

We are happy that the reviewer sees the novelty of this part of the paper. We have chosen to improve Figure 7 by directly following the referee's suggestions. Firstly, we now have graphs (Fig. 7A and C) to "...show the dynamics of the 4 different genes with respect to time (and not only the bifurcation diagram)...". Secondly, we have also changed the bifurcation diagrams to make them clearer: The scale of the axes is kept fixed throughout so that it is easier to see how the full nullclines move, and how they alter the fixed points over time (with none of them appearing out of nowhere). Additionally, our choice of axes has made it easier for the reader to compare them directly to the equivalent bifurcation diagrams of the mono-functional modules (H1 and D3).

- as far as I can see, the behaviour of this emergent networks bears strong resemblance with a recent « geometric » analysis by Corson and Siggia, PNAS 2012, where they show using pure geometries in phase space how one can change direction of the trajectory of the system and get to a new basin of attraction on the *C. elegans* vulva example. The problem is very close in spirit because changes of basins is due to cell-cell interaction as well, but Corson and Siggia are also predictive of actual data. This raises two questions: 1> how are the topology and geometry in phase space presented here comparable to this previous work ? and 2> does that mean that we should focus on phase space geometry rather than network topology, as proposed by Corson and Siggia ?

We thank the reviewer for pointing out that we had not put the "geometric" approach into its proper historical context. As the reviewer is aware, there is a sparse but long history of visualizing cell type specification using the simple graphical tools of dynamical systems theory – in particular, geometric phase portraits or bifurcation diagrams. One of the earliest and clearest was J. Slack (1991) who used a phase portrait to show the relationship between a molecular circuit and the resulting attractors, separatrix, and the variety of potential trajectories for a hypothetical fate choice between epithelium and neural cell types. Much later, one of the first to apply this approach to real quantitative data was Huang et al (2007) who explicitly visualized the differentiation choice between erythroid and myeloid cell types using expression data of the two transcription factors PU.1 and GATA1 which form a mutual-inhibition circuit. They were able to infer the basins of attraction for this system, and to define the classic pitchfork bifurcation as an explanation of switching from the intermediate progenitor cell type to either of the two differentiated cell types. More recent papers have continued to explore this dynamical systems approach in a variety of contexts – such as Macia et al 2009, Manu et al 2009, Corson & Siggia 2012, Munteanu et al. 2014, Verd 2014 and others.

We had probably taken it too much for granted that this geometric approach is widely understood to be the correct and useful way to analyze these systems. However, following the reviewers' comments, we have now improved the phrasing of these sections and added these important references.

Thank you again for submitting your work to Molecular Systems Biology. We have now heard back from reviewer #1 who was asked to evaluate your study. As you will see below, s/he is satisfied with the modifications made and raises only two remaining minor concerns, which we would ask you to address in a minor revision.

REFEREE REPORT

Reviewer #1:

The reviewers have satisfactorily addressed all of my concerns. I am also very happy with the change in title and narrative that arose from a comment made by the other reviewer.

Additionally, I think that the grouping of earlier work based on the four themes of "mutually-compatible functions," "multi-stable circuits," "altering circuit structure," and "multifunctional circuits" is particularly helpful, not only to put this paper into context, but also as a reference for the field. For this reason, I think it's important to address two minor concerns:

(1) The authors emphasize that the two functions they consider are qualitatively different, and they are. But they use this point to contrast with some of the papers in the "multi-stable circuits" grouping, saying that the functions of such circuits "do not correspond to qualitatively distinct dynamical behaviors --- they are all stable point attractors." I respectfully disagree. In these circuits, it could also be the case that one function is a point attractor and another is a cyclic attractor. If the authors agree that these are qualitatively distinct dynamical behaviors, then I would ask them to modify the above statement accordingly.

(2) I am thankful that the authors discuss how "multi-stable" functions can also be thought of as simply having a single function --- that of classification. However, I think it is worth noting that the circuits studied by the authors could also be thought of as a single decision-making function. They map the state of the input node onto one of two possibilities: lateral induction or lateral inhibition.

But these are small concerns. This is a fantastic paper and I am happy to endorse its publication.

2nd Revision - authors' response

16 March 2017

Reviewer #1:

The reviewers have satisfactorily addressed all of my concerns. I am also very happy with the change in title and narrative that arose from a comment made by the other reviewer.

Additionally, I think that the grouping of earlier work based on the four themes of "mutually-compatible functions," "multi-stable circuits," "altering circuit structure," and "multifunctional circuits" is particularly helpful, not only to put this paper into context, but also as a reference for the field. For this reason, I think it's important to address two minor concerns:

(1) The authors emphasize that the two functions they consider are qualitatively different, and they are. But they use this point to contrast with some of the papers in the "multi-stable circuits" grouping, saying that the functions of such circuits "do not correspond to qualitatively distinct dynamical behaviors --- they are all stable point attractors." I respectfully disagree. In these circuits, it could also be the case that one function is a point attractor and another is a cyclic attractor. If the authors agree that these are qualitatively distinct dynamical behaviors, then I would ask them to modify the above statement accordingly.

The reviewer is correct here. We have altered the sentence to read as follows:

“However, the different end-states in these examples (whether Boolean or continuous) are typically stable point attractors, which do not correspond to qualitatively distinct dynamical behaviors (although cyclic attractors are also possible in these systems).”

(2) I am thankful that the authors discuss how "multi-stable" functions can also be thought of as simply having a single function --- that of classification. However, I think it is worth noting that the circuits studied by the authors could also be thought of as a single decision-making function. They map the state of the input node onto one of two possibilities: lateral induction or lateral inhibition.

We fully appreciate the reviewer’s point, but respectfully we feel that there is a meaningful difference between our 2 different patterning functions, and a decision-making system. We have thus altered a sentence in the section on Multi-functional circuits as follows: “These examples tend to be quite simple dynamical systems, nevertheless this type of multi-functionality is closer in spirit to the general biological phenomenon of pleiotropy because the alternative functions are not simply alternative decision states (which need not be qualitatively distinct), but instead they directly embody the distinct dynamical behaviors of two different biological functions.”

But these are small concerns. This is a fantastic paper and I am happy to endorse its publication.

We are extremely happy to read these comments – thanks very much for a productive peer-review process!

3rd Editorial Decision

20 March 2017

Thank you again for sending us your revised manuscript. We are now satisfied with the modifications made and I am pleased to inform you that your paper has been accepted for publication.

James Sharpe

Molecular Systems Biology

16-7347